# TabKDE: Simple and Scalable Tabular Data Generation with Kernel Density Estimates

## Abstract

Tabular data generation considers a large table with multiple columns – each column comprised of numerical, categorical, or sometimes ordinal values. The goal is to produce new rows for the table that replicate the distribution of rows from the original data – without just copying those initial rows. The last 3 years has seen enormous progress on this problem, mostly using computational expensive methods that employ one-hot encoding, VAEs, and diffusion.

This paper describes a new approach to the problem of tabular data generation. By employing copula transformations and modeling the distribution as a kernel density estimate we can nearly match the accuracy and privacy-preservation achievements of the previous methods, but with almost no training time. Our method is very scalable, and can be run on data sets orders of magnitude larger than prior art on a simple laptop. Moreover, because we employ kernel density estimates, we can store the model as a coreset of the original data – we believe the first for generative modeling – and as a result, require significantly less space as well. Our code is available here: `http://github.com/tabkde/tabkde-main`

## 1 Introduction

Tabular data is a fundamental format in many domains, including finance, healthcare, and social sciences, and has seen much recent attention (Fonseca and Bacao, 2023; Assefa et al., 2021; Hernandez et al., 2022; Ouyang et al., 2023), focusing on challenges in scalability and accuracy in its diverse structural characteristics (Xu et al., 2019; Borisov et al., 2023; Liu et al., 2023). Unlike image or text data, which follow well-defined spatial or sequential relationships, tabular data consists of mixture of varied features that may be numerical, categorical, or ordinal. This heterogeneity poses difficulties in modeling feature dependencies and joint distributions effectively. Traditional generative approaches, such as GANs and VAEs, have been applied with mixed success, and may need to paired with careful preprocessing and one-hot encoding, which can lead to an explosion in dimensionality and loss of information (Xu et al., 2019; Zhang et al., 2024). Moreover, adversarial training in GANs can be unstable (Arjovsky and Bottou, 2017), while VAEs may struggle to generate realistic samples due to overly simplistic latent space assumptions (Dai and Wipf, 2019).

Copula-based data generators Patki et al. (2016); Majdara and Nooshabadi (2020) provide another approach towards transforming differently structured and scaled columns into common format. The synthetic data vault (SDV) Patki et al. (2016) includes generative modeling through a variety of approaches including low-rank modeling, GANs, and vine-copula Meyer et al. (2021). This framework can achieve improved fidelity, but sometimes at a heavy computational expense.

Diffusion models have recently emerged as powerful generative frameworks, demonstrating impressive performance in domains such as image synthesis and molecular generation (Ho et al., 2020; Rombach et al., 2022; Dhariwal and Nichol, 2021; Morehead and Cheng, 2024; Luo et al., 2024). These models operate by progressively transforming noise into structured data through a learned denoising process. Recent advances, such as TabDDPM (Kotelnikov et al., 2023), TabSYN (Zhang et al., 2024), and TabDiff Shi et al. (2025) have made significant progress in adapting diffusion to the tabular setting. TabDDPM (Kotelnikov et al., 2023) applies a diffusion model directly to tabular data, effectively capturing complex distributions but requiring a high number of sampling steps. TabSYN (Zhang et al., 2024) introduced a latent-space diffusion approach (similar to stable diffusion model approach (Rombach et al., 2022)): it first encodes categorical features with a one-hot

encoding, then invokes a VAE to map to a structured representation before applying a diffusion model. This approach has demonstrated remarkable improvements in synthetic data quality, outperforming previous methods in terms of statistical fidelity. TABDIFF (Shi et al., 2025) extends this by applying a discrete state-space diffusion for the categorical features.

**Challenges with the synthetic tabular data generation.** Ultimately, effective tabular data generation needs to over come three challenges. First, it should achieve **high accuracy** in how the distribution of data it generates aligns with heldout data along marginal, pairwise correlations, and full joint distributional measurements. Most prior methods without diffusion are not able to hit high levels of accuracy. Second, it should **preserve privacy** of the test data; it cannot just generate synthetic data too similar to the data it was trained on. For instance, SMOTE (Chawla et al., 2002) while successful in other metrics, often re-generates training data, or very close to it. Third, it should be **scalable and efficient**; that is, it should be able to easily handle very large training sets, and – more challengingly – data with many categories. Methods based on one-hot encoding like TABSYN can run out of memory with large numbers of categories, and diffusion-based approaches can be relatively slow to train. *No prior method achieves all three desiderata*; see Table 1.

## 1.1 OUR CONTRIBUTION

We propose a new approach to tabular data generation, TABKDE, that achieves all three desiderata; see Table 1. Notably, it only uses classic tools (carefully assembled): copula transformation, covariance estimation, kernel density estimation. We argue this simplicity improves the interpretability, and we demonstrate (in Section 3) that while nearly-matching SOTA accuracy and privacy it significantly reduces the computational cost in training and generation. It has the following specific advantages.

- **Scalability.** It is more scalable and efficient than methods (like TABDDPM Kotelnikov et al. (2023) and TABSYN Zhang et al. (2024)) which rely on one-hot encoding and need to train an expensive diffusion model. These falter on datasets with many categories.

- **Preserving Privacy.** TABKDE preserves privacy while not requiring extensive training. In contrast SMOTE Chawla et al. (2002), which also does not have a training step, often generates data too close to the data it was trained on.

- **Coreset for Tabular Data Generation.** For the first time, we construct a coreset for tabular data generation. By mapping data into a space where kernel density estimates are applicable, we can apply coresets for KDEs which compactly (and sub-linearly in training size) represents the generative process.

Table 1: Comparison of popular tabular data synthesis methods across key criteria.

| Method | citation | Scalable | Accurate | Private |
|---|---|:---:|:---:|:---:|
| SMOTE | (Chawla et al., 2002) | ✓ | ✓ | x |
| GReaT | (Borisov et al., 2023) | x | x | x |
| GOGGLE | (Liu et al., 2023) | x | x | x |
| CoDi | (Lee et al., 2023) | x | x | x |
| STaSy | (Kim et al., 2023) | x | x | x |
| TabDDPM | (Kotelnikov et al., 2023) | x | ✓ | ✓ |
| TABSYN | (Zhang et al., 2024) | x | ✓ | ✓ |
| CORETABKDE | coreset variant | ✓ | ✓ | ✓ |
| TABKDE | main contribution | ✓ | ✓ | ✓ |

**Overview of our approach: TABKDE.** We follow the general three step paradigm of Rombach et al. (2022), applied to tabular data. **(1) Encoding** converts the input into a standardized continuous representation. After this step each of categorical, ordinal, and numerical features are then represented in the same continuous format. **(2) Embedding into a Distance-Aware Latent Space** uses a continuous mapping into another continuous space where now Euclidean distance between objects is representative of how similar they are. **(3) Generative Modeling** maps the discrete distribution of training data in the latent space to a continuous distribution from which we can sample from. The

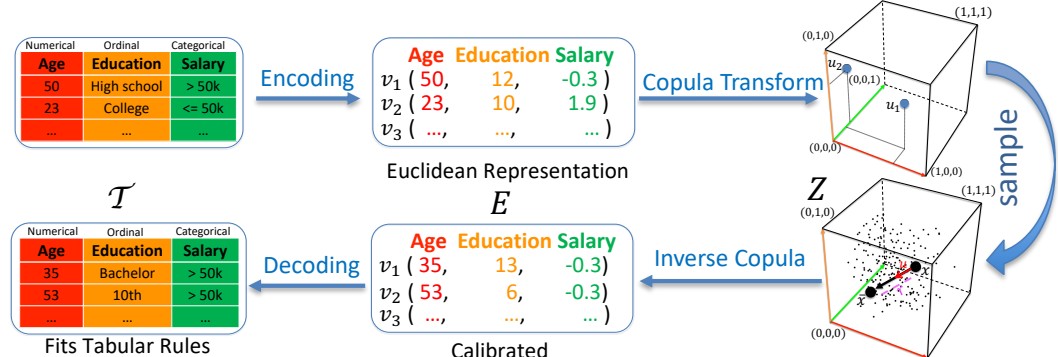

Figure 1: An overview of the proposed TABKDE.

samples are then made in the latent space, inverted to the encoded format, and decoded to be in the format of the input table.

Figure 1 illustrates how we implement the general framework; notably we avoid one-hot encoding (which can blow up memory requirements), VAEs and diffusion (which can be slow to train), to achieve a scalable method which can still achieve high accuracy and retain privacy.

Our method TABKDE first converts all features—numerical, ordinal, and categorical—into a unified numerical format; importantly, it does so in a careful way, so each column in the table is represented as a single numerical value. This is followed by a copula-based transformation that maps the data into a unit hypercube for easier (marginal) density estimation. Here its covariance is also calibrated. Then the generative process uses Kernel Density Estimation (KDE) modeling to represent and sample from a distribution. The only training is in learning the shape of the kernel to match the distance to closest record. Note that sampling from a KDE is simple, in that it needs only to choose a data point, a (covariance scaled) direction, and an offset distance. Then generated samples are mapped back by inverting the copula transform, and decoded. One key additional step is performed to ensure all samples are within the margins identified by the copula, otherwise partial resampling is performed.

## 2 TABKDE ALGORITHM

We consider a tabular dataset $\mathcal{T} = \{X_1, X_2, \ldots, X_m\}$, where each row $X_i$ represents an independent and identically distributed (i.i.d.) sample from an unknown joint distribution $P(X)$. Each row $X_i = (x_1, x_2, \ldots, x_d)$ is a $d$-dimensional vector where each feature $x_j$ belongs to one of three categories: Numerical (Num), Ordinal (Ord), and Categorical (Cat). Each numerical is in $\mathbb{R}$, but both categorical and ordinal features come from a discrete domain; the difference being that ordinal features have a specified ordering (e.g., grades $A > B > C$).

The *key innovation* of TABKDE is in the careful combination of copula mapping to latent space and the KDE-based generative modeling approach.

**Copula Latent Space Mapping:** It uses a copula-based transformation of the tabular dataset $\mathcal{T}$ into a latent representation $Z$ that lies within the continuous $[0,1]^d$, allowing precise control over the domain of the marginal distributions. This dimension and margin preservation critically leverages principal vector guided encoding of categorical features. Then the covariance $\Sigma$ of data in this space allows for modeling the directional variation of the data within these bounds.

**KDE Generative Modeling:** Here we use a KDE model to allow for a non-parametric complex distributional model. We simply sample by choosing a training data point in the latent space $z \sim \mathsf{Unif}(Z)$, and then choose a random direction $u$ proportional to the covariance $\Sigma$. Then generate:

$$z' \leftarrow z + ur$$

where $r$ is a scalar controlling the amount of deviation from the training data. In particular, we select $r$ at randomly from a learned distribution from the training data estimating the distance to the closest point. One more idea is needed, we disallow points outside $[0,1]^d$ to respect the original column marginals. To handle this, we keep coordinates within $[0,1]^d$, and regenerate the others iteratively.

## 2.1 Encoding of Tabular Features: $\mathcal{T} \to E$

For tabular data $\mathcal{T} = \{X_1, \ldots, X_n\}$ with $d$ coordinates, we first encode each row $X_i$ into a space $E \subset \mathbb{R}^d$. Importantly the space $E$ has one dimension for each column in $\mathcal{T}$. Numerical values are left as they are, and categorical values are assigned values $1, 2, 3, \ldots$ for the rank of the ordinal category.

For categorical data we avoid one-hot encoding, and instead apply PRINCIPALGUIDEDENCODING. It first computes a vector $u \in \mathbb{R}^{d'}$, the top principal vector of the $d' \leq d$ numerical features; this captures the largest mode of variation among the numerical data. Then for each categorical dimension $C_j$ (for $j \in [d]$) it uses $u$ to numerically encode each category. That is, for each category $c \in C_j$ it considers all rows $X_i$ with $c$ in coordinate $j$; call this set $\mathcal{T}_{j=c}$. Then for numerical parts $x_i^{\mathsf{num}} \in \mathbb{R}^{d'}$ of each $X_i \in \mathcal{T}_{j=c}$, it finds their average numerical value along direction $u$; that is $v_{j=c} = \frac{1}{|\mathcal{T}_{j=c}|} \sum_{X_i \in \mathcal{T}_{j=c}} \langle u, x_i^{\mathsf{num}} \rangle$. The discrete category $c$ is replaced with that average value $v_{j=c}$ in the space $E$. These categorical dimension can be detokenized by randomly rounding to one of the two nearest values proportional to how close it is.

## 2.2 Map to Numerical Latent Space: $E \to Z$

Next we map to a latent space $Z$ where the distances measure closeness between objects. Our representation will have two aspects. First we will ensure $Z = [0,1]^d$, so each coordinate is continuous value between $0$ and $1$. We do this with a copula transform (Patki et al., 2016) where among the encoded training data $E_i \in E$, each coordinate $E_{i,j}$ is assigned its value in the empirical CDF. So $Z_{i,j}$ is the fraction of $E_{i',j} \leq E_{i,j}$. We store the sorted order of the values $E_{i,j}$ so we can invert this.

Second, after have built $Z$ via a copula map in each coordinate, we then compute sample covariance $\Sigma$ of $Z$. This induces a Mahalanobis distance, and $\Sigma$ will be important for generative sampling.

## 2.3 Learning Distance to Closest Record (DCR)

In generative sampling, one often computes the *distance to closest record (DCR)* (Mateo-Sanz et al., 2004; Steier et al., 2025) to evaluate how similar a synthetic record $s$ is to a real one from a set $Z$.

$$\text{DCR}(s, Z) = \min_{z \in Z} \|s - z\|$$

A DCR of $0$ may indicate an identical match, posing a significant privacy risk. We can comparing DCR values from synthetic data to both training $Z_T$ and holdout $Z_H$ datasets. Ideally, for synthetic data $S$, DCR distributions $\{\text{DCR}(s, Z_T)\}_{s \in S}$ and $\{\text{DCR}(s, Z_H)\}_{s \in S}$ should heavily overlap, showing that synthetic data reflects general patterns rather than replicating specific records.

As part of our generative process, we learn this distribution $\{\text{DCR}(s, Z)\}_{s \in S}$ where $Z$ is the training data in the copula space. Then we can generate synthetic data to mirror this scale of variation. We repeatedly randomly split the training data, and computes the DCR distribution between the two splits. Then we fit a simple mixture of $k$ Gaussians to this distribution; choosing $k$ (from $1, \ldots, 10$) using Bayesian Information Criterion.

## 2.4 Tabular Kernel Density Estimation: $Z \to$ Sample

A *kernel density estimation (KDE)* is a continuous estimate of a probability density function built by smoothing finite data samples with a kernel function $K$ (often Gaussian) and bandwidth $h$. For $n$ data points $X = \{x_1, \ldots, x_n\} \sim P$, if we appropriately adjust $h$ as $n$ grows, then the KDE defined $\text{KDE}_X(x) = \frac{1}{n} \sum_{i=1}^n K((x - x_i)/h)$ will converge to $P$ (Silverman, 1986; Scott, 2015). Moreover, we can generate synthetic data (in a manner that approaches the unknown distribution $P$) by drawing a random point $x_i$ and then adding an offset defined by $K$ and $h$.

**Sampling from KDE with DCR kernel.** In our TABKDE method we adapt this sampling from a KDE of the data in a few subtle ways. First, instead of a Gaussian, we use a kernel with offset radius $r$ matching a learned DCR distribution. Second, instead of selecting the offset direction $u$ uniformly, we draw it proportional to the learned covariance $\Sigma$. These two modifications are sketched in Algorithm 1 with a single sample in latent space generated via Algorithm 2.

**Algorithm 1** SIMPLEKDE($\mathcal{T}$)

1: $Z \in [0,1]^{n \times d} \Leftarrow$ Copula-Transform($\mathcal{T}$)
2: $\Sigma \leftarrow$ Covariance($Z$)
3: Estimate empirical DCR distribution $f$
4: **for** $i = 1, \ldots, m$:
5:     $z_i' =$ SAMPLEKDE($Z, f, \Sigma$)
6:     $y_j \leftarrow$ INVERSECOPULA($z_j'$)
7: **return** $Y = \{y_1, \ldots, y_m\}$

**Algorithm 2** SAMPLEKDE($Z, f, \Sigma$)

1: Uniformly sample $z_i \in Z$
2: Sample radius $r > 0$ from $f$
3: Sample $v \sim \mathcal{N}(0, \Sigma)$, set $u = \frac{v}{\|v\|}$
4: **return** $z' \leftarrow z_i + r \cdot u$

However, the SIMPLEKDE algorithm does not explicitly control the support of the marginals, which is critical for tabular data generation. The copula-transformed representation, however, embeds the data within the unit hypercube, which allows us to control how far a perturbed sampled point $x$ can deviate without violating marginal support. We now introduce a more refined rejection-sampling heuristic (Alg. 3: TABKDE) that effectively enforces these boundary constraints.

**Algorithm 4** SAMPLEKDE-ITERATIVE($Z, f, \Sigma$)

1: Uniformly sample $z_i \in Z$
2: Sample radius $r > 0$ from $f$
3: Sample $v \sim \mathcal{N}(0, \Sigma)$, set $u = \frac{v}{\|v\|}$
4: $z' \leftarrow z_i + r \cdot u$
5: **While** $\{j : z_j' \notin [0,1]\} \neq \varnothing$:
6:     $J \leftarrow \{j : z_j' \notin [0,1]\}$
7:     Sample $v' \sim \mathcal{N}(0, \Sigma)$, set $w = \frac{v'}{\|v'\|}$
8:     $s \leftarrow \frac{\|(u_k)_{k \in J}\|}{\|(w_k)_{k \in J}\|}$
9:     $u_j \leftarrow s \cdot w_j$ for each $j \in J$
10:    $z' \leftarrow z_i + r \cdot u$
11: **return** $z'$

**Algorithm 3** TABKDE($\mathcal{T}$)

1: $Z \in [0,1]^{n \times d} \Leftarrow$ Copula-Transform($\mathcal{T}$)
2: $\Sigma \leftarrow$ Covariance($Z$)
3: Estimate empirical DCR distribution $f$
4: **for** $i = 1, \ldots, m$:
5:     $z_i' =$ SAMPLEKDE-ITERATIVE($Z, f, \Sigma$)
6:     $y_j \leftarrow$ INVERSECOPULA($z_j'$
7: **return** $Y = \{y_1, \ldots, y_m\}$

TABKDE differs from SIMPLEKDE only in line 5, where it uses the boundary-aware SAMPLEKDE-ITERATIVE (Alg 4) instead of the simpler SAMPLEKDE. This modified sampler checks for violations of the unit hypercube boundaries and regenerates out-of-bound coordinates. If a valid point cannot be obtained after a fixed number of attempts, the sample is discarded, and the process restarts. This mechanism guarantees that all accepted samples lie within the latent space $[0,1]^d$.

## 2.5 CORESETS FOR GENERATIVE TABULAR DATA MODELING

A *coreset* (Phillips, 2016) is a compact, weighted set of points that provides a close approximation to the full dataset for a specific downstream task. In the context of KDEs, a coreset serves to approximate the full KDE using significantly fewer, strategically chosen, representative points.

Our proposed TABKDE framework employs the full $\text{KDE}_Z$ to generate samples from the Copula latent representation $Z \subset [0,1]^{n \times d}$ of the tabular data $\mathcal{T}$. To approximate $\text{KDE}_Z(\cdot)$ using a coreset, we define $\tilde{\text{KDE}}_\Theta(\cdot)$ with $\Theta$ comprised of a small set of learnable coreset points $Q = \{q_1, \ldots, q_m\}$ and their corresponding non-negative weights $W = \{\omega_1, \ldots, \omega_m\}$, constrained such that $\sum_{i=1}^m \omega_i = 1$. The approximated density function is $\tilde{\text{KDE}}_\Theta(z) = \sum_{i=1}^m \omega_i K\left(\frac{z - q_i}{h}\right)$. It is known (Phillips and Tai, 2020) that a sample $Q \sim Z$ of $m = O((1/\varepsilon^2) \log(1/\delta))$ points, and uniform weights already ensures a strong $L_\infty$ coreset approximation that $\|\text{KDE}_Z - \text{KDE}_Q\|_\infty \leq \varepsilon$ with probability at least $1 - \delta$. For the Gaussian kernel (used here) and a fixed bandwidth $h$, this bound is independent of dimension $d$.

Moreover, this can be used as a starting point for an optimized coreset, over the locations $Q$ and their weights $W$ to minimize the empirical $L_2$ via SGD as $\mathbb{E}_{z \sim \text{Unif}([0,1]^d)}\left[\left(\tilde{\text{KDE}}_\Theta(z) - \text{KDE}_Z(z)\right)^2\right]$. This optimized version can potentially better preserve key distributional features, such as modes, spread, and overall shape. Moreover, because we are not replicating the training data, it can reduce the risk of overfitting to the data or leaking its private attributes. We call this method CORETABKDE; and RANDCORETABKDE only samples but does not optimize. We use coresets of size $m = 5000$.

Table 2: Runtime comparison of Tabsyn, TabKDE, and SMOTE models across individual datasets on laptop. The IBM dataset is excluded from the average row.

| Dataset | TABSYN | | | | SMOTE | TABKDE | |
|---------|-----------|-------------|-------------|--------|--------------|---------|--------|
| | VAE Train | Diff. Train | Total Train | Sample | Train+Sample | Train | Sample |
| Adult | 6h 35m 43s | 2h 6m 31s | 8h 43m 19s | 1m 5s | 4s | 44s | 20s |
| Default | 6h 32m 3s | 2h 2m 16s | 8h 34m 59s | 40s | 2s | 59s | 17s |
| Shoppers | 3h 57m 42s | 0h 55m 32s | 4h 53m 32s | 18s | 3s | 17s | 5s |
| Magic | 3h 51m 7s | 1h 21m 27s | 5h 13m 0s | 26s | 5s | 19s | 7s |
| Beijing | 5h 31m 57s | 1h 57m 44s | 7h 30m 35s | 54s | 2s | 35s | 16s |
| News | 14h 34m 15s | 2h 8m 59s | 16h 44m 11s | 57s | 4s | 6m 2s | 54s |
| **Average** | **6h 50m 27s** | **1h 45m 24s** | **8h 36m 36s** | **43s** | **3s** | **1m 29s** | **19s** |
| IBM | OOM | OOM | OOM | OOM | OOM | 10m 21s | 6m 4s |

## 3 EXPERIMENTAL RESULTS

Our experiments are conducted on the six tabular datasets from UCI Machine Learning Repository[1] (Adult, Default, Shoppers, Magic, Beijing, News) with between 12K and 49K rows and a mixture of 11 and 48 dimensions, a mixture of mostly numerical and categorical. We also use an IBM dataset[2] which is significantly larger; it has about 176K rows and 14 dimensions, with 5 ordinal ones, and a total of over 37K total categories. See Appendix B.1 for more detail.

**Baselines.** We compare our proposed TABKDE method with several popular baselines, including SMOTE (Chawla et al., 2002), GReaT (Borisov et al., 2023), CoDi (Lee et al., 2023), TabD-DPM (Kotelnikov et al., 2023), TABSYN (Zhang et al., 2024), and TABDIFF (Shi et al., 2025); some comparisons and other methods are deferred to Appendix B.2 for space. We also consider several hybrid models that mix elements of TABKDE with the encoding choices. Notably, in COPULADIFF we first use our COPULAMAPPING to embed data into a latent space, train a diffusion model there. Broader comparisons with other copula-based methods are in Appendix H.

### 3.1 SCALABILITY AND EFFICIENCY

We measure the scalability and efficiency on both the training time, as well as the sample generation time; sample generation measures time for the full synthetic set – the same size as training set.

Table 3: Average GPU timing

| Method | Train (s) | Sample (s) |
|--------|-----------|------------|
| GReaT | 17112.4 | 251.2 |
| CoDi | 18487.6 | 11.8 |
| TabDDPM | 2771.4 | 70.8 |
| TABSYN | 1297.8 | 8.4 |
| TABKDE | 39.2 | 39.0 |

We first compare against TABSYN and SMOTE on a laptop using only CPU (2021 Apple 14" MacBook Pro; M1 Pro chip). Table 2 shows that the simple SMOTE algorithm is faster than TABKDE, but the training time of TABKDE is orders of magnitude faster than TABSYN (about 90 seconds to about 8.5 hours). Appendix C shows other baselines have run times in the ballpark of TABSYN. Moreover, both TABSYN and SMOTE run out of memory on the IBM data set since they try to one-hot encode 37K categories, while TABKDE still completes in under 20 minutes.

We compare GPU runtime (NVIDIA RTX A5000; 24GB memory; max power 230W) in Table 3 over the average training and sampling time on Adult, Default, Shoppers, and Magic datasets. TABKDE is still orders of magnitude faster in training, and while other methods can improve upon TABKDE (our code not optimized for GPU) in sampling, this cost is dominated by the training time.

### 3.2 ACCURACY

In this section, we evaluate the quality of the generated synthetic data using three criteria: (1) marginal distribution alignment, (2) pairwise correlation matching, and (3) finally global alignment between synthetic and hold-out distributions is compared by how well a classifier can separate the distributions.

---

[1] https://archive.ics.uci.edu/datasets

[2] https://www.kaggle.com/code/yichenzhang1226/ibm-credit-card-fraud-detection-eda-random-forest

**Marginal distribution alignment.** When evaluating synthetic tabular data, the marginal distribution alignment score assesses how closely each individual column matches its real-data distribution represented by train data. Following what was done in the TABSYN paper, we calculate the Kolmogorov–Smirnov (KS) distance for numerical attributes in Num and the Total Variation distance for categorical and ordinal attributes in Cat and Ord. Table 4 presents, for each dataset, the average marginal alignment errors across all features for each method.

Table 4: Marginal distribution alignment error; lower is better. In parentheses denotes ratio relative to the smallest value. Baseline values taken from Zhang et al. (2024)

| Method | Adult | Default | Shoppers | Magic | Beijing | News | Average |
|---|---|---|---|---|---|---|---|
| SMOTE | 1.63 (2.55) | 1.70 (1.49) | 2.66 (2.16) | 1.37 (1.93) | 2.10 (1.62) | 5.47 (3.18) | 2.49 (1.75) |
| GReaT | 12.12 (18.94) | 19.94 (17.49) | 14.51 (11.80) | 16.16 (22.76) | 8.25 (6.35) | – | 14.20 (10.00) |
| CoDi | 21.38 (33.41) | 15.77 (13.82) | 31.84 (25.89) | 11.56 (16.28) | 16.94 (13.03) | 32.27 (18.78) | 21.63 (15.23) |
| TabDDPM | 1.75 (2.73) | 1.57 (1.38) | 2.72 (2.21) | 1.01 (1.42) | 1.30 (1.00) | 78.75 (45.83) | 14.52 (10.23) |
| TabSYN (Our reproduced) | 0.64 (1.00) | 1.14 (1.00) | 1.23 (1.00) | 0.98 (1.38) | 2.79 (2.15) | 1.72 (1.00) | 1.42 (1.00) |
| COPULADIFF | 2.01 (3.14) | 1.47 (1.29) | 2.47 (2.01) | 0.94 (1.32) | 2.13 (1.64) | 2.44 (1.42) | 1.91 (1.35) |
| RANDCORETABKDE | 1.61 (2.52) | 1.76 (1.54) | 2.54 (2.07) | 1.01 (1.42) | 1.70 (1.31) | 2.59 (1.51) | 1.87 (3.48) |
| CORETABKDE | 3.63 (5.67) | 3.29 (2.89) | 3.23 (2.63) | 1.08 (1.52) | 3.20 (2.46) | 2.87 (1.67) | 2.88 (2.03) |
| TABKDE | 1.56 (2.44) | 1.55 (1.36) | 2.44 (1.98) | 0.78 (1.1) | 1.37 (1.05) | 2.52 (1.47) | 1.70 (1.2) |

Figure 2 provides a visual comparison between some representative selected real marginal distributions and those generated by TABSYN (orange) and TABKDE (green) against the real data distributions (blue). We observe that TABKDE and TABSYN visually match the distributions well, both are about the same. In particular, on numerical data TABKDE seems to do better on more uniform distributions whereas TABSYN does better on spiky ones.

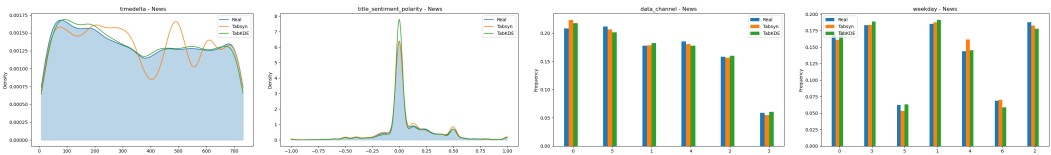

Figure 2: Marginals comparison between real data (blue), TABKDE (green), and TABSYN (orange). Representative numerical and categorical data from News dataset.

**Pairwise correlation alignment.** We next measure pairwise correlation between columns. For numerical-numerical pairs, we can use standard Pearson correlations. For pairs that involve categorical or ordinal feature (as in Zhang et al. (2024)) we use contingency-table total variation distances. In both metrics, smaller error values indicate that the synthetic table is more faithful to the original data. Table 5 presents, for each dataset, the average pairwise correlation alignment errors across all features for each method. A heatmap visualization of the divergence between the pairwise correlations in the real and synthetic data is presented in Figure 3; see more in Appendix D.2. We observe that TABKDE has better pairwise correlation alignment than all methods except TABSYN, and is comparable to SMOTE; both TABKDE and SMOTE have about 2× the correlation discrepancy as TABSYN.

Table 5: Pairwise correlation alignment error; Lower values is better. In parentheses is the ratio relative to the smallest value. Baselines values taken from Zhang et al. (2024).

| Method | Adult | Default | Shoppers | Magic | Beijing | News | Average |
|---|---|---|---|---|---|---|---|
| SMOTE(our reproduction) | 4.3 (2.67) | 11.54 (5.11) | 3.68 (1.47) | 1.88 (2.29) | 3.3 (1.22) | 1.67 (1.25) | 4.39 (1.99) |
| GReaT | 17.59 (10.93) | 70.02 (30.98) | 45.16 (18.06) | 10.23 (12.48) | 59.6 (21.99) | – | 40.52 (18.33) |
| CoDi | 22.49 (13.98) | 68.41 (30.27) | 17.78 (7.11) | 6.53 (7.97) | 7.07 (2.61) | 11.1 (8.28) | 22.23 (10.06) |
| TabDDPM | 3.01 (1.87) | 4.89 (2.16) | 6.61 (2.64) | 1.7 (2.07) | 2.71 (1.00) | 13.16 (9.82) | 5.34 (2.42) |
| TABSYN (our reproduction) | 1.61 (1.00) | 2.26 (1.00) | 2.5 (1.00) | 0.82 (1.00) | 4.7 (1.73) | 1.34 (1.00) | 2.21 (1.00) |
| COPULADIFF | 4.61 (2.86) | 3.29 (1.46) | 5.3 (2.12) | 1.72 (2.10) | 4.5 (1.66) | 2.1 (1.57) | 3.59 (1.62) |
| RANDCORETABKDE | 3.93 (2.44) | 13.66 (6.04) | 4.27 (1.71) | 4.76 (5.80) | 4.05 (1.49) | 2.61 (1.95) | 5.46 (2.47) |
| CORETABKDE | 6.3 (3.91) | 9.91 (4.39) | 5.77 (2.31) | 2.18 (2.66) | 5.86 (2.16) | 2.82 (2.10) | 5.47 (2.48) |
| TABKDE | 4.51 (2.80) | 9.93 (4.40) | 4.31 (1.72) | 2.72 (3.32) | 3.74 (1.38) | 2.83 (2.11) | 4.67 (2.11) |

We also compare against some variants on the larger IBM dataset in Figure 6. Recall that on our laptop

Table 6: Accuracy on IBM.

| Method | Marginal | Pairwise | Reduced |
|---|---|---|---|
| TABSYN | 16.99 | 40.42 | no |
| COPULADIFF | 6.81 | 29.42 | no |
| COPULADIFF | 3.59 | 22.59 | yes |
| CORETABKDE | 5.25 | 23.93 | yes |
| TABKDE | 3.58 | 25.29 | yes |

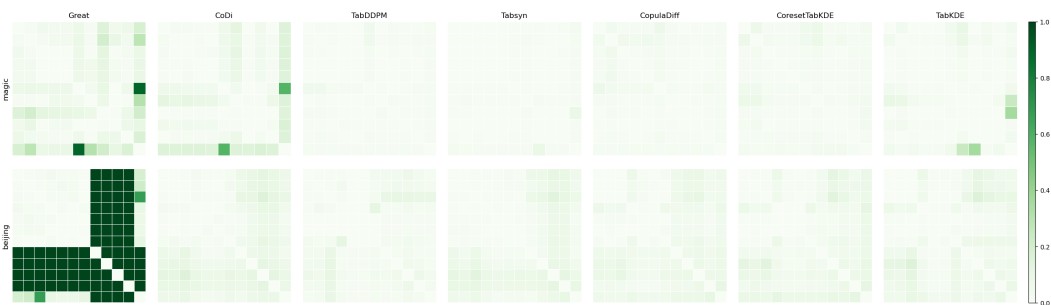

Figure 3: Representative Pairwise correlation divergence heatmaps for Magic and Beijing datasets.

CPU, neither TABSYN or SMOTE can run on this data set – they both run out of memory. For TABKDE we apply a data reduction modeling by first reducing out 3 strongly correlated categories (i.e., state and city depend on zipcode), then inferring them in the decoding step; details in Appendix B.1.

Recall also that TABKDE (about 10 minutes training time on CPU; 40 seconds on GPU) was much faster than either of COPULADIFF (about 15 hours on CPU), or TABSYN (OOM on CPU, about 20 minutes on GPU). The average pairwise correlation alignment error for TABSYN (no reduction) is 40.42%, while for TABKDE and COPULADIFF (with reduction) is 25.29% and 22.59%.

**Global Distribution Alignment.** Synthetic data should be able to take the place of real data, letting us train models on it for downstream prediction tasks and have indistinguishable performance. We assess by building a classifier to attempt to distinguish between the synthetic data and a split of data held out from the training process. We use logistic regression in Table 7 and random forest in Appendix D. We quantify this as a classifier two-sample test (C2ST) as provided by SDMetrics; larger values closer to 1 are better.

We observe that TABKDE is roughly the same as SMOTE with 0.93 and only bested by TABSYN which has about 0.97. Other baselines achieve 0.79 (TabDDPM) or below 0.66.

Table 7: C2ST Scores; larger is better. Baselines taken from Zhang et al. (2024)

| Method | Adult | Default | Shoppers | Magic | Beijing | News | Average |
|---|---|---|---|---|---|---|---|
| Smote | 0.9212 | 0.9332 | 0.9107 | 0.9803 | 0.9972 | 0.8633 | 0.9334 |
| GReaT | 0.5376 | 0.4710 | 0.4285 | 0.4326 | 0.6893 | — | 0.5118 |
| CoDi | 0.2077 | 0.4595 | 0.2784 | 0.7206 | 0.7177 | 0.0201 | 0.4007 |
| TabDDPM | 0.9755 | 0.9712 | 0.8349 | 0.9998 | 0.9513 | 0.0002 | 0.7888 |
| TABSYN(Our reproduction) | 0.9949 | 0.9804 | 0.9699 | 0.9893 | 0.9268 | 0.9584 | 0.9699 |
| COPULADIFF | 0.8557 | 0.9798 | 0.8665 | 0.9914 | 0.9576 | 0.9793 | 0.9384 |
| RANDCORETABKDE | 0.9215 | 0.9570 | 0.8757 | 0.9921 | 0.9503 | 0.8901 | 0.9311 |
| CORETABKDE | 0.8254 | 0.873 | 0.8462 | 0.9864 | 0.8924 | 0.8643 | 0.8813 |
| TABKDE | 0.9219 | 0.9579 | 0.9161 | 1.0000 | 0.9514 | 0.8819 | 0.9382 |

## 3.3 PRIVACY

Finally, we evaluate how well we can preserve the privacy of the training set in synthetic data generation through DCR. For each synthetic data point generated, we compute the distance to training and held-out data. Ideally these distributions should be indistinguishable.

First in Table 8 we calculate the *DCR score*, which is the percentage of synthetic data closer to training data than held-out; we would like this to be close to 50%. This was proposed for TABDIFF (Shi et al., 2025), and we show their results in the top half of the table, and also show SMOTE, TABSYN, and our methods following their code below the line. We see most diffusion methods (including our COPULADIFF) can consistently achieve below 52%. Our main method TABKDE

obtains an average DCR score of about 58%, which is servicable. SMOTE has an average DCR score of 95% indicating that it reveals significant information (if not replicating) the training data.

Table 8: The DCR score for synthetic data sample comparing training to held out data. A value nearer to 50% is considered ideal. Baseline values taken from (Shi et al., 2025).

| Method | Adult | Default | Shoppers | Beijing | News | Average |
|---|---|---|---|---|---|---|
| Smote (Our reproduction) | 91.18 (1.82) | 91.46 (1.82) | 96.76 (1.92) | 100.00 (1.99) | 99.00 (1.96) | 95.68 (1.89) |
| CoDi | 49.92 (1.00) | 51.82 (1.03) | 51.06 (1.02) | 50.87 (1.01) | 50.79 (1.00) | 50.89 (1.01) |
| TabDDPM | 51.14 (1.02) | 52.15 (1.04) | 63.23 (1.26) | 80.11 (1.59) | 79.31 (1.57) | 65.19 (1.29) |
| TABDIFF | 50.10 (1.00) | 51.11 (1.02) | 50.24 (1.00) | 50.50 (1.00) | 51.04 (1.01) | 50.60 (1.00) |
| TABSYN (Our reproduction) | 51.33 (1.02) | 51.61 (1.02) | 51.99 (1.06) | 53.20 (1.00) | 50.76 (1.01) | 51.65 (1.02) |
| COPULADIFF | 50.34 (1.01) | 50.96 (1.01) | 50.72 (1.01) | 50.29 (1.00) | 53.00 (1.05) | 51.06 (1.01) |
| RANDCORETABKDE | 62.30 (1.24) | 63.09 (1.26) | 58.91 (1.17) | 63.50 (1.26) | 55.59 (1.10) | 60.68 (1.20) |
| CORETABKDE | 52.59 (1.05) | 54.11 (1.08) | 55.04 (1.1) | 51.17 (1.03) | 52.00 (1.02) | 52.98 (1.05) |
| TABKDE | 62.23 (1.24) | 63.46 (1.26) | 58.80 (1.17) | 54.24 (1.11) | 54.54 (1.08) | 58.55 (1.16) |

The *DCR score is an imperfect measure of privacy*, since there may be heldout data nearly as close as the training data to a synthetic point. Indeed TABKDE is inspired by differential privacy, and the accepted comparison is the ratio of likelihoods, not the count of which one is maximum. So another, albeit less quantitative, evaluation considers the DCR distribution of synthetic data measured to training versus heldout data. Figure 4 shows distribution to training (blue) versus to heldout (red) for representative data on Beijing and News; more in Appendix E. We observe that for TABKDE, TABSYN, and CORETABKDE these distributions are multi-modal, but still match almost perfectly. On the other hand SMOTE has a very different distribution, and the synthetic to train (blue) is always much smaller (typically very close to 0), indicating it may often reproduce the training data.

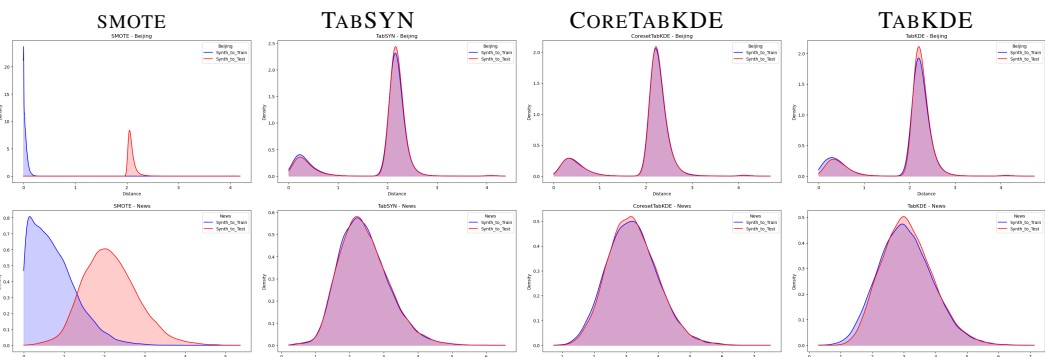

Figure 4: Privacy comparison based on DCR distributions for synthetic to training data (blue) and synthetic to held-out data (red). First row Beijing and second row News.

**Coreset methods.** The coreset methods CORETABKDE and RANDCORETABKDE work nearly as well as TABKDE in terms of accuracy, and use a fraction of the space. Surprisingly, RAND-CORETABKDE often has better accuracy than CORETABKDE and is of course faster since it does not require the optimization step. However, notably, CORETABKDE has a much improved DCR score for privacy (of about 53%), so provides a way to address that measure within the TABKDE framework. See Appendix F for a longer discussion.

### 3.4 CONCLUSION AND LIMITATIONS

We introduce a new approach for tabular data generation, built with a careful combination of only classic techniques like copula transforms and KDEs. It is the first to demonstrate high scalability, accuracy, and privacy. While its DCR privacy score is not quite as strong as other methods, this can be improved with slight accuracy trade-offs through coresets. It handles, but also requires, a mix of numerical and categorical features as is common in tabular data.

**LLM Disclosure.** In this paper, the use of LLMs is restricted to enhancing grammar and making partial rewording adjustments.

**Reproducibility Statement.** We clearly describe the full TabKDE algorithm and the other hybrid models, including encoding, copula transformation, KDE sampling, and coreset construction (Sections 2 and Appendix A). Section 3 outlines the datasets used, evaluation metrics, and experimental setups, while Tables 2 and 3 include compute resources and runtime information. Additional implementation details are provided in Appendices B–F. Together, these components enable reproduction of the key results, even independently of the code. Our code is also anonymously available here: `http://github.com/tabkde/tabkde-main`

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

# A  TABKDE ALGORITHM

The TABKDE algorithm is a simple, scalable, and privacy-aware method for generating high-fidelity synthetic tabular data. It implements the general framework (see Section 1) for tabular generation; the key innovation is in the mapping to latent space and generative modeling sets. These steps are simple and efficient while satisfying our desiderata. We overview them here:

**TabKDE Latent Space Mapping:** This is accomplished in two parts. The first step is a copula-based transformation of the tabular dataset $\mathcal{T}$ into a latent representation $Z$ that lies within the continuous space $[0,1]^d$, allowing precise control over the domain of the marginal distributions. Second is estimating the covariance $\Sigma$ of data in this space, implicitly defining a Mahalanobis distance which captures the similarity within this latent domain.

**TabKDE Generative Modeling:** Here we use a KDE model to allow for a non-parametric complex distributional model. We simply sample by choosing a training data point in the latent space $z \sim \mathsf{Unif}(Z)$, and then chose a random direction $u$ scaled by the covariance $\Sigma$. Then we generate a new point

$$z' \leftarrow z + ur$$

where $r$ is a scalar amount controlling the amount of deviation from the training data. In particular, we select $r$ at randomly from a learned distribution from the training data estimating the distance to the closest point. One more idea is needed, we want to disallow points outside $[0,1]^d$ to respect the original column marginals. To handle this, we keep coordinates in $[0,1]^d$, and regenerate the others iteratively.

We next describe all aspects in more detail.

## A.1  ENCODING OF TABULAR FEATURES: $\mathcal{T} \to E$

Let $\mathcal{T} = \{X_1, \ldots, X_n\}$ be a real data set. Before transforming tabular data into a latent space, it is essential to convert all feature types into a unified numerical format $\mathcal{T} \mapsto E \in \mathbb{R}^{n \times d}$, suitable for further processing. Note that the new representation $E$ has $n$ rows (one for each row of the table $\mathcal{T}$), and more importantly $d$ columns (one for each column of the table $\mathcal{T}$). This means, do not use one-hot-encoding, and this will be essential for our representation and sampling from the latent space to ensure marginal properties of each data column.

In this section, we describe how we handle numerical, ordinal, and categorical features through encoding strategies designed to preserve basic structural relationships and facilitate meaningful downstream transformations.

**Numerical and Ordinal Mapping.**  We keep the numerical features unchanged, ensuring that their original values are retained without any transformation. Ordinal features are simply converted to consecutive integers $1, 2, \ldots,$ that preserve their inherent order. This ensures that the ordinal relationships between categories are preserved while converting them into numerical representations.

**Categorical Mapping.**  For categorical features, we use a data-driven approach to map each unique category to a continuous space based on the statistical properties of the numerical features in the data.

Recall that $\mathsf{Cat} = \{d_2 + 1, \ldots, d\}$ denotes the set of indices corresponding to categorical features. Additionally, assume that for each $j \in \mathsf{Cat}$, the number of unique categories in the $j$-th feature is given by $|C_j| = k_j$. Define $Z \in \mathbb{R}^{n \times d_1}$ as the matrix obtained by selecting only the numerical features from the dataset, and let $u$ represent its principal direction obtained using Principal Component Analysis (PCA). Let $j \in \mathsf{Cat}$ be an arbitrary categorical feature with unique category set $C_j = \{c_1, \ldots, c_{k_j}\}$. For each category $c \in C_j$, we identify the corresponding row indices in $D$, defined as

$$I_{j,c} = \{i \in [n] \mid (X_i)_j = c\}.$$

We then assign each category $c$ the value

$$v_{j,c} = \frac{1}{|I_{j,c}|} \sum_{i \in I_{j,c}} u_i,$$

This value represents the average of the principal direction components $u_i$ for all instances where the $j$-th categorical feature takes the value $c$. This process is summarized in the following algorithm.

---

**Algorithm 5** PRINCIPALGUIDEDENCODING($\mathcal{T}$, Cat, Num)

---

1: Compute $u$ as the principal direction of $X_{\text{Num}}$; which contains only numerical features in $\mathcal{T}$.
2: **for** each category $c \in C_j$, in each categorical feature $j \in$ Cat:

$$v_{j,c} \leftarrow \frac{1}{|I_{j,c}|} \sum_{i \in I_{j,c}} u_i, \quad \text{where } I_c = \{i \in [n] \mid (X_i)_j = c\}$$

3: **for** each data $(X_i)_j = c \in C_j$, for each categorical feature $j \in$ Cat: $\quad (E_i)_j \leftarrow v_{j,c}$

---

### A.2 MAP TO NUMERICAL LATENT SPACE: $E \rightarrow Z$

Notably, after the initial encoding step (Subsection A.1), all features in the dataset are converted into numerical values, resulting in a representation that lies in a subset of $\mathbb{R}^d$. Our goal is to further transform this representation into a continuous latent space within the unit hypercube $[0,1]^d$, where the dependencies between features are preserved, and the data is appropriately normalized to ensure that all numerical features contribute equally, making it well-suited for downstream tasks such as sampling and density estimation.

The algorithm outlined below–MAPTOLATENTSPACE–provides a high-level overview of this transformation process. It combines ordinal encoding, a structure-aware encoding of categorical features, and a copula-based normalization of all features. While this procedure is presented here in full, each of its core components will be introduced and discussed in detail in the subsequent sections.

---

**Algorithm 6** MAPTOLATENTSPACE($\mathcal{T}$, Num, Cat, Ord)

---

1: Mapped each ordinal feature to integers reflecting its natural order.
2: Encoded categorical features by PRINCIPALGUIDEDENCODING($\mathcal{T}$, Cat, Num)
3: Concatenate numerical and the transformed ordinal and encoded categorical features to obtain $E$
4: Convert the encoded data $E$ into $Z = $ COPULAMAPPING($E$) $\in [0,1]^{n \times d}$.
5: **return** $Z$

---

Before exploring the details of this mapping, we first introduce foundational concepts from the copula method—a well-established statistical technique that separates marginal distributions from the dependency structure in multivariate data.

**Introduction to Copula Transformation.** In many real-world datasets, variables exhibit complex dependencies, making it challenging to model their joint distribution directly. Copula method provides a powerful statistical tool to decouple the dependency structure from the marginal distributions, allowing for more flexible data transformations. The copula method invertibly transforms a dataset $E \in \mathbb{R}^{n \times d}$, consisting of $d$-dimensional features, into a new representation $Z \in [0,1]^{n \times d}$ and as a result, each individual record in $Z$ lies in $[0,1]^d$, and the marginal distributions of $Z$ are uniform over the interval $[0,1]$. We next examine the underlying mechanism by which this transformation is achieved.

1. **Copula Forward Transformation (Mapping $E$ to $Z$):** For each dimension $j$ (where $j = 1, \ldots, d$), we compute the empirical cumulative distribution function (ECDF) of the $j$-th feature:

$$\hat{F}_j(x) = \Pr(\text{value of } j\text{-th coordinate} \leq x)$$

$$= \frac{1}{n} \sum_{i=1}^{n} \mathbb{I}(x_{ij} \leq x)$$

where $\mathbb{I}(x_{ij} \leq x)$ is an indicator function that equals 1 if $x_{ij} \leq x$, and 0 otherwise. In summary, $\hat{F}_j(x)$ represents the proportion of observations in the dataset whose $j$-th

coordinates are less than or equal to $x$. Each coordinate value $x_{ij}$ is then transformed into a uniform representation:

$$z_{ij} = \hat{F}_j(x_{ij}) \tag{1}$$

This ensures that each feature is uniformly mapped into the interval $[0, 1]$, producing a dataset $Z$ that follows a uniform distribution for each of its marginals while maintaining the dependency structure of $X$.

---

**Algorithm 7** COPULAMAPPING($E$)

---

1: For each feature $j$, compute empirical CDF value $\hat{F}_j(x_{i,j})$ for $D_j = \{x_{1j}, \ldots, x_{nj}\}$.
2: Set $z_i = (z_{i,1}, \ldots, z_{i,d})$, where $z_{i,j} = \hat{F}_j(x_{i,j})$
3: **return** $Z = \{z_1, \ldots, z_n\}$

---

Using the Copula transformation, we effectively *standardize* the data set into a unit hyper-cube, making it more suitable for density estimation, sampling, and synthetic data generation. Furthermore, this method allows for *dependency-preserving transformations*, ensuring that the statistical relationships between variables are retained even when synthetic data is produced.

2. **Copula Inverse Transformation (Mapping $Z$ back to $E$):** It is key that we store the ECDF $\hat{F}$, because we need to be able to invert it. Its inverse cumulative distribution function (quantile function) $\hat{F}^{-1}$ is defined as

$$\hat{F}^{-1}(q) = \inf\{x \mid \hat{F}(x) \geq q\} \qquad \text{for any value} \qquad q \in [0, 1].$$

Given $z = (z_1, \ldots, z_d) \in [0, 1]^d$, each feature $j$ can mapped back to its initial numerical representation using the inverse cumulative distribution function $\hat{F}_j^{-1}(\cdot)$.

**Decoding.** We can also decode the output $E$ to the structure in the table. For an ordinal or categorical feature $j$, we apply probabilistic rounding to the two nearest categories (in the initial numerical embedding), with the probabilities proportional to their distance to $p_j$. Also, for numerical features, the value is reconstructed (up to the appropriate precision) by interpolating between the two closest values (in the original distribution) to $p_j$, with the interpolation weights determined by their distance from $p_j$. This step ensures that the generated samples maintain the same marginal distributions as the original dataset.

---

**Algorithm 8** INVERSEECDF($z = (p_1, \ldots, p_d), \{\hat{F}_i : i \in [d]\}$)

---

1: **for** each $j \in [d]$:
2:     **if** $\min(\{z_{ij}\}_{i=1}^n) \geq p_j$ or $\max(\{z_{ij}\}_{i=1}^n) \leq p_j$ **then**
3:         **Return** $\min(\{x_{ij}\}_{i=1}^n)$ or $\max(\{x_{ij}\}_{i=1}^n)$ respectively
4:     Let $z_{i_1} < z_{i_2}$ be consecutively ordered points so that $p \in [z_{i_1}, z_{i_2}]$
5:     **if** $j \in \mathsf{Ord}$ or $j \in \mathsf{Cat}$, **then**
6:         **Return** $x_j = x_{i_1}$ with probability $\frac{|p_j - z_{i_1}|}{|z_{i_1} - z_{i_2}|}$ and otherwise $x_{i_2}$.
7:     **elseif** $j \in \mathsf{Num}$, **then**
8:         **Return**

$$x = x_{i_2} + \frac{|p_j - z_{i_2}|}{|z_{i_1} - z_{i_2}|}(x_{i_1} - x_{i_2}) \tag{2}$$

---

## A.3   LEARNING DISTANCE TO CLOSEST RECORD (DCR)

A central use of synthetic data is as a proxy for private personal data. So it is paramount to ensure that the synthetic data process is not leaking too much information about the original data. A common measurement of this is **Distance to Closest Record (DCR)** (Mateo-Sanz et al., 2004; Steier et al., 2025) which evaluates how similar a synthetic record $x_s$ is to a real one from a set $D$. It is formally defined for an appropriate distance metric $\mathtt{d}$ as:

$$\text{DCR}(x_s, D) = \min_{x_r \in D} \text{d}(x_s, x_r) \tag{3}$$

A DCR of $0$ indicates an identical match, posing a significant privacy risk. Comparing DCR values between synthetic data and both training $D_T$ and holdout $D_H$ datasets helps assess privacy. If synthetic records are much closer to the training data, it suggests the model may be memorizing real data. Ideally, for synthetic data $S$, DCR distributions $\{\text{DCR}(x_s, D_T)\}_{x_s \in S}$ and $\{\text{DCR}(x_s, D_H)\}_{x_s \in S}$ **should heavily overlap**, showing that synthetic data reflects general patterns rather than replicating specific records.

As part of our generative process, we learn this distribution in the copula latent embedding $Z$ using *Euclidean distance*. Then we can generate synthetic data to mirror this scale of variation. We repeatedly randomly split the training data $Z$, and computes the DCR distribution between the two splits (see EMPIRICALDCR; Alg. 9). Then it fits a simple mixture of Gaussians model to this distribution; using Bayesian Information Criterion (BIC)[3], we select the best model for $k = 1, \ldots, 10$ as the number of components.

---

**Algorithm 9** EMPIRICALDCR($Z$): Estimating the Empirical DCR Distribution

---

1: Initialize $L = []$
2: **for** $i = 1, \ldots, T$ **do**
3:     Partition $Z$ into two random equal-sized subsets $Z_1$ and $Z_2$.
4:     **for** each $z_2 \in Z_2$ **do**
5:         Compute the minimum distance between $z_2$ and the records in $Z_1$.
6:         Add this distance to $L$.
7: Fit a mixture of $k$ Gaussian components to $L$.

---

### A.4 TABULAR KERNEL DENSITY ESTIMATION: $Z \rightarrow$ SAMPLE

KDE (Kernel Density Estimation) is a non-parametric method used to estimate the probability density function (PDF) of a continuous random variable by smoothing finite data points with a kernel function (typically Gaussian). Its accuracy depends on bandwidth selection and data availability (Silverman, 1986; Scott, 2015). It can also be used to generate synthetic data by fitting a KDE model to the existing dataset and drawing samples from it. We now formally define KDE.

Assuming that $X = \{x_1, x_2, \ldots, x_n\}$ is a dataset in $\mathbb{R}^d$, the Kernel Density Estimation (KDE) is given by:

$$\hat{f}(x) \propto \frac{1}{n} \sum_{i=1}^{n} K\left(\frac{x - x_i}{h}\right)$$

where:

- $\hat{f}(x)$ is the estimated likelihood at point $x$,
- $K(\cdot)$ is a centrally-symmetric kernel function (e.g., Gaussian kernel),
- $h > 0$ is the bandwidth parameter controlling smoothness.

**Sampling from KDE.** To sample from a $\text{KDE}_X$, we simply sample a point $x \in X$, and then sample a point nearby proportional to the kernel likelihood. For example, with a isotropic Gaussian kernel, using this approach, a synthetic data point is generated as $x \sim \mathcal{N}(x_i, \frac{h}{2}I);\ x_i \sim X$. However, we do not use a Gaussian kernel, as this is not adaptive to the DCR distribution – which may be multi-modal. So we use our EMPIRICALDCR($Z$) estimate to define out kernel. Still, this approach may raise a couple of concerns:

---

[3]https://scikit-learn.org/stable/modules/generated/sklearn.mixture.GaussianMixture.html#sklearn.mixture.GaussianMixture.bic

- At first glance, one may worry that this approach does not guarantee preservation of DCR, since it might use our kernel $K(x_i, \cdot)$ to generate a point nearby $x_i$ that lands too close to another $x' \in X$. However, this is not observed to be an issue, as generation in high-dimensional space makes it highly unlikely to produce points close to other points in the training data (see Subsection 2.3).

- Second, the sampled point $x$ may fall outside the convex hull of the dataset $X$, potentially resulting in unrealistic data generation. We address this by using the sample covariance $\Sigma$ to guide the perturbation direction. And Moreover, some form of extrapolation in this sense is probably necessary and unavoidable (Balestriero et al., 2021), and we beleive desirable. Yet, violating the marginals associated with individual table columns, we find, can distort distributions (see Subsection G.1). This issue is addressed by generalizing SIMPLEKDE to a more advanced method, TABKDE, which controls for this.

To resolve the first issue, we estimate the DCR distribution using Algorithm 9 (EMPIRICALDCR) and leverage it to strategically perturb the sampled point $x_i$. We summarize this approach by the following algorithms: Algorithm 10 SIMPLEKDE($\mathcal{T}$), which iteratively calls Algorithm 11 SAMPLEKDE($Z, f, \Sigma$) using the copula-transformed data $Z$, its estimated DCR distribution $f$, and its estimated covariance $\Sigma$.

---

**Algorithm 10** SIMPLEKDE($\mathcal{T}$)

---

1: Transform table $\mathcal{T}$ into $Z \in [0,1]^{n \times d}$ as $Z \leftarrow \text{COPULATRANSFORM}(\mathcal{T}, \mathsf{Num}, \mathsf{Cat}, \mathsf{Ord})$
2: $\Sigma \leftarrow \text{Covariance}(Z)$
3: Estimate empirical DCR distribution $f = \text{EMPIRICALDCR}(Z)$
4: **for** $i = 1, \ldots, m$:
5: $\quad z'_i = \text{SAMPLEKDE}(Z, f, \Sigma)$
6: $\quad y_j \leftarrow \text{INVERSEECDF}(z'_j, j\text{-th feature type}, F_j)$
7: **return** $Y = \{y_1, \ldots, y_m\}$

---

**Algorithm 11** SAMPLEKDE($Z, f, \Sigma$)

---

1: Uniformly sample $z_i \in Z$
2: Sample radius $r > 0$ from $f$
3: Sample $v \sim \mathcal{N}(0, \Sigma)$, set $u = \frac{v}{\|v\|}$
4: **return** $z' \leftarrow z_i + r \cdot u$

---

As discussed in Subsection G.1, the SIMPLEKDE algorithm does not explicitly control the support of the marginals and, in particular, does not fully address the second challenge outlined earlier. The copula-transformed representation, however, embeds the data within the unit hypercube, which allows us to control how far a perturbed sampled point $x$ can deviate without violating marginal support. We now introduce a more refined rejection-sampling heuristic (Alg. 12: TABKDE) that effectively enforces these boundary constraints.

---

**Algorithm 12** TABKDE(X)

---

1: Transform table $\mathcal{T}$ into $Z \in [0,1]^{n \times d}$ as $Z \leftarrow \text{COPULATRANSFORM}(\mathcal{T}, \mathsf{Num}, \mathsf{Cat}, \mathsf{Ord})$
2: $\Sigma \leftarrow \text{Covariance}(Z)$
3: Estimate empirical DCR distribution $f = \text{EMPIRICALDCR}(Z)$
4: **for** $i = 1, \ldots, m$:
5: $\quad z'_i = \text{SAMPLEKDE-ITERATIVE}(Z, f, \Sigma)$
6: $\quad y_i = \text{INVERSECOPULA}(z'_i)$
7: **return** $Y = \{y_1, \ldots, y_m\}$

---

TABKDE differs from SIMPLEKDE only in the sampling step at line 5, where it uses the boundary-aware SAMPLEKDE-ITERATIVE instead of the simpler SAMPLEKDE. This modified sampler actively

checks for violations of the unit hypercube boundaries and iteratively adjusts any out-of-bound coordinates. If a valid point cannot be obtained after a fixed number of attempts, the sample is discarded, and the process restarts. This mechanism guarantees that all accepted samples lie within the latent space $[0, 1]^d$.

---

**Algorithm 13** SAMPLEKDE-ITERATIVE(Z)

---

1: Uniformly sample $z_i \in Z$
2: Sample radius $r > 0$ from $f$
3: Sample $v \sim \mathcal{N}(0, \Sigma)$, set $u = \frac{v}{\|v\|}$
4: $z' \leftarrow z_i + r \cdot u$
5: **While** $\{j : z'_j \notin [0, 1]\} \neq \varnothing$:
6:      $J \leftarrow \{j : z'_j \notin [0, 1]\}$
7:      Sample $v' \sim \mathcal{N}(0, \Sigma)$, set $w = \frac{v'}{\|v'\|}$
8:      $s \leftarrow \frac{\|(u_k)_{k \in J}\|}{\|(w_k)_{k \in J}\|}$
9:      $u_j \leftarrow s \cdot w_j$ for each $j \in J$
10:     $z' \leftarrow z_i + r \cdot u$
11: **return** $z'$

---

# B EXPERIMENTAL SETUP AND DATA

## B.1 DATASETS

Our experiments are conducted on the six tabular datasets from UCI Machine Learning Repository[4] (Adult, Default, Shoppers, Magic, Beijing, News ) used in TABSYN (Zhang et al., 2024), along with the IBM dataset[5] which is significantly larger. In all they covering a wide range of domains for tabular data. These datasets include a mix of numerical and categorical features and vary in the number of points, feature types, and task types (classification or regression), making them well-suited for evaluating the generalizability of synthetic data generation methods. A few of the categorical features can be interpreted as ordinal; but outside the IBM dataset, we simply treat them as categorical. We summarize their traits in Table 9.

Table 9: Dataset statistics. **Num** denotes the number of numerical columns, **Cat** the number of categorical columns, **Ord** the number of ordinal features, and **Sum Cat** the total number of unique categories across all categorical and ordinal columns. Ordinal features can be treated as categorical features by disregarding their inherent order; note (∗) that we do this for the Adult and Default datasets. For the IBM dataset, we randomly select two 200k subsamples to serve as the training and testing sets; we ensured that the test set contains no categorical values unseen in the training set. In IBM data, we treat "*Year*", "*Month*", "*Day*", "*Time*", "*Zip*" features as ordinal. (†) In TABKDE, SIMPLE-KDE, COPULADIFF and CORETABKDE, for Beijing dataset, we treat the features "Is", "Ir", and "Iws" as categorical, and for the Shoppers dataset, we apply the same treatment to the features "SpecialDay", "ProductRelated", and "Informational". For all of these features, the ratio of unique values in the training set to the total number of data points is very low, with the majority of occurrences concentrated on a single value.

| Dataset | Total | Train | Test | Num | Cat | Ord | Sum Cat | Task |
|---------|-------|-------|------|-----|-----|-----|---------|------|
| Adult | 48,842 | 32,561 | 16,281 | 6 | 7 | 2* | 120 | Classification |
| Default | 30,000 | 27,000 | 3,000 | 14 | 9 | 1* | 79 | Classification |
| Shoppers† | 12,330 | 11,097 | 1,233 | 10 | 8 | 0 | 67 | Classification |
| Magic | 19,019 | 17,117 | 1,902 | 10 | 1 | 0 | 2 | Classification |
| Beijing† | 41,757 | 37,581 | 4,176 | 7 | 5 | 0 | 76 | Regression |
| News | 39,644 | 35,679 | 3,965 | 46 | 2 | 0 | 13 | Regression |
| IBM | 341,675 | 176,221 | 165,454 | 1 | 8 | 5 | 37,721 | Classification |

---

[4] https://archive.ics.uci.edu/datasets
[5] https://www.kaggle.com/code/yichenzhang1226/ibm-credit-card-fraud-detection-eda-random-forest

**Split of the Data.** We consider two ways to split data into test and train set. The numbers in Table 9 reflects the splits done by TABSYN, which we maintain for direct comparison. This split was not even in size, partially to ensure there were no categories in the Test split which were not present in the Train split. When we do not directly compare to results in the Zhang et al. (2024), we use a different random and even split.

Again, following the TABSYN paper (Zhang et al., 2024), the target column is treated as either numerical or categorical based on the task type: it is considered categorical for classification tasks and numerical otherwise. For the Machine Learning Efficiency experiments, each dataset is divided into training, validation, and testing sets. For the Adult dataset, we use its official testing set, while the original training set is further split into training and validation sets with an 8:1 ratio. All other datasets are split into training, validation, and testing sets using an 8:1:1 ratio with a fixed random seed.

To get the IBM dataset to run, we needed to leverage the ordinal representation in variables Year, Month, Day, Time, and Zip. We also identify that the Merchant State and Merchant City are very strongly correlated with Zip (the zip code), and since these categories are often quite rare, we applied another modeling (called *reduced modeling*) to improve the generation. We put Zip in the generative model, but not Merchant City or Merchant State. Then once we generate a Zip, we predict the the Merchant City and Merchant State. We use the same process with the correlated MCC and Merchant Name; we put MCC in the generative model, and use the outcome to predict Merchant Name.

### B.2 BASELINES

We compare our proposed TABKDE method with several popular baselines, including SMOTE (Chawla et al., 2002), CTGAN(Xu et al., 2019), TVAE (Xu et al., 2019), GReaT (Borisov et al., 2023), GOGGLE (Liu et al., 2023), CoDi (Lee et al., 2023), STaSy (Kim et al., 2023), Tab-DDPM (Kotelnikov et al., 2023), TABSYN (Zhang et al., 2024), and TABDIFF (Shi et al., 2025)[6]. Through this comparison, we demonstrate that TABKDE provides a simpler, faster, and more scalable alternative for generating realistic synthetic tabular data, without significantly compromising on quality or privacy.

By abstracting to the general framework (outlined in Section 1 ) we are able to also consider several hybrid models that mix elements of TABKDE with the encoding choices, the VAE method, or the diffusion-driven generation made popular through TABSYN and others. We specifically consider

- **COPULADIFF:** We first use our COPULAMAPPING (Alg. 7) to embed data into a latent space, train a Diffusion model there, and then map the generated samples back to the original space.
- **VAETABKDE:** This model trains a VAE to embed data into numerical space (as in TAB-SYN), then applies the Copula and KDE methods to generate samples, which are then mapped back to the original tabular format.
- **VAESIMPLEKDE:** VAESIMPLEKDE differs from VAETABKDE only in the sampling step, analogous to how TABKDE differs from SIMPLE-KDE.
- **PGE-TABSYN:** In the method we replace one-hot encoding with the encoding outlined in Subsection A.1 (see the first three steps in MAPTOLATENTSPACE Alg. 6), which tokenizes the tabular data into the space $E$ before applying VAE and Diffusion models.

### B.3 EVALUATION

To evaluate our synthetic data generation method, we focus on three main objectives including **1) Scalability**, **2) Accuracy**, and **3) Privacy**. In terms of Efficiency, we measure and compare the training and sampling time required by each model across various datasets. For Accuracy, we assess how well the synthetic data captures (1) the ground truth marginals for each column individually, (2)

---

[6]We have not yet been able to reproduce all results for TABDIFF, but they report (Shi et al., 2025) very similar performance to TABSYN in terms of accuracy, efficiency, and privacy – although with small, but noticeable improvement on categorical data. This follows since they build directly on most of the framework of TABSYN except for using a discrete diffusion on the categorical parts. For the comparison to TABKDE and its relatives, TABSYN's performance should serve as a good empirical representative.

correlations for pairs of columns, (3) the entire distribution using machine learning efficiency, and (4) the balance between fidelity and coverage using $\alpha$-Precision and $\beta$-Recall. For Privacy, we use the Distance to Closest Records (DCR) metric to assess privacy protection by measuring how similar the synthetic data is to the training/test data sets.

## C  SCALABILITY AND EFFICIENCY

We measure the scalability and efficiency on both the training time, as well as the sample generation time; this second part is the generation of a sample the same size as the training set. As shown in Tables 10 and 11, we see in most existing methods the time is almost entirely dominated by the training aspect. However, TABKDE the training and sampling are more comparable because the training time is so much lower.

All our experiments, unless otherwise specified, were conducted using only the CPU of a 2021 Apple MacBook Pro (14-inch), equipped with an Apple M1 Pro chip. This device features an 8-core CPU (comprising 6 performance cores and 2 efficiency cores) and 16 GB of unified memory. Table 10 presents a comparison of the training times between TABSYN and SMOTE against our proposed TABKDE, demonstrating that our method is highly computationally efficient and can be effectively executed on a standard consumer-grade laptop. TABSYN and SMOTE both run out of memory on the IBM data set, this is primarily because it has a huge number of categories, and these methods rely on one-hot encoding, which blows up the dimensionality into a 37,733-dimensional space. The memory inefficient one-hot encoding is standard in many modern models. SMOTE requires this dimensionality to identify the $k$ nearest neighbors, which becomes highly inefficient in such a high-dimensional space (see Table 9 for dataset details).

In contrast, our proposed tokenization method, PRINCIPALGUIDEDENCODING (Algorithm 5), transforms tabular data into a numerical format with a fixed dimensionality equal to the original number of features (14 in this case, compared to 37722 with one-hot encoding), providing a far more efficient representation.

Table 10: Runtime comparison of Tabsyn, TabKDE, and SMOTE models across individual datasets on laptop. The IBM dataset is excluded from the average row.

| Dataset | TABSYN | | | | SMOTE | TABKDE | |
|---------|--------|--|--|--|-------|--------|--|
| | VAE Train | Diff. Train | Total Train | Sample | Train+Sample | Train | Sample |
| Adult | 6h 35m 43s | 2h 6m 31s | 8h 43m 19s | 1m 5s | 4s | 44s | 20s |
| Default | 6h 32m 3s | 2h 2m 16s | 8h 34m 59s | 40s | 2s | 59s | 17s |
| Shoppers | 3h 57m 42s | 0h 55m 32s | 4h 53m 32s | 18s | 3s | 17s | 5s |
| Magic | 3h 51m 7s | 1h 21m 27s | 5h 13m 0s | 26s | 5s | 19s | 7s |
| Beijing | 5h 31m 57s | 1h 57m 44s | 7h 30m 35s | 54s | 2s | 35s | 16s |
| News | 14h 34m 15s | 2h 8m 59s | 16h 44m 11s | 57s | 4s | 6m 2s | 54s |
| **Average** | **6h 50m 27s** | **1h 45m 24s** | **8h 36m 36s** | **43s** | **3s** | **1m 29s** | **19s** |
| IBM | OOM | OOM | OOM | OOM | OOM | 10m 21s | 6m 4s |

The baseline methods in the TABSYN (Zhang et al., 2024) paper were evaluated on the Adult dataset using an NVIDIA RTX 4090 GPU with 24 GB of memory, as shown in Table 11. In contrast, our experiments—including those for TABSYN and the proposed TABKDE—were conducted entirely on significantly less powerful hardware. Despite this substantial difference in computational resources, TABKDE demonstrates superior efficiency. As shown in Table 10, it achieves an average training time of only 1 minute and 29 seconds, and a sampling time of just 19 seconds across six benchmark datasets. This is considerably faster than the TABSYN model, which requires more than 8.5 hours of training on average. These results highlight that TABKDE not only nearly matches the performance of more complex models (see Section D) but also does so at a fraction of the computational cost, making it highly suitable for deployment on standard consumer-grade machines without the need for specialized accelerators.

As previously noted, running TABSYN on the IBM dataset is infeasible given our standard computational resources. This limitation arises mainly due to using one-hot encoding—results in a

Table 11: Training and sampling times for baseline methods on the Adult dataset, evaluated using an NVIDIA RTX 4090 GPU with 24 GB of memory (adapted from the TABSYN Zhang et al. (2024) paper).

| Method | Training Time | Sampling Time |
|---|---|---|
| CTGAN | 17 min 10 s | 0.86 s |
| TVAE | 5 min 53 s | 0.51 s |
| GOGGLE | 1 h 34 min | 5.34 s |
| GReaT | 2 h 27 min | 2 min 19 s |
| STaSy | 38 min 3 s | 8.94 s |
| CoDi | 2 h 56 min | 4.62 s |
| TabDDPM | 17 min 11 s | 28.92 s |
| TABSYN | 40 min 22 s | 1.78 s |

37,733-dimensional feature space. However, we can alternatively use encoding scheme introduced in the preprocessing steps of TABKDE. Accordingly, we apply both COPULADIFF and PGE-TABSYN to the IBM dataset. Training COPULADIFF requires 15 hours and 7 minutes, while PGE-TABSYN demands over 40 hours in total—25 hours and 56 minutes for the VAE and 14 hours and 39 minutes for the diffusion stage. As shown in Table 10, TABKDE only takes about 10 minutes of training time. This comparison further highlights TABKDE 's advantage in scalability over more resource-intensive methods.

For a direct comparison, we evaluate TABKDE alongside the baselines TABSYN, TabDDPM, CoDi, and GReat on the Adult, Default, Shoppers, and Magic datasets, using an NVIDIA RTX A5000 GPU with 24GB of memory and a maximum power draw of 230W, under the same experimental settings as TABSYN. See Tables 12 and 13 and Figure 5 for details. TABKDE is orders of magnitude faster in training, but on par with others in sampling time – we generate samples sequentially, and did not optimize for the GPU.

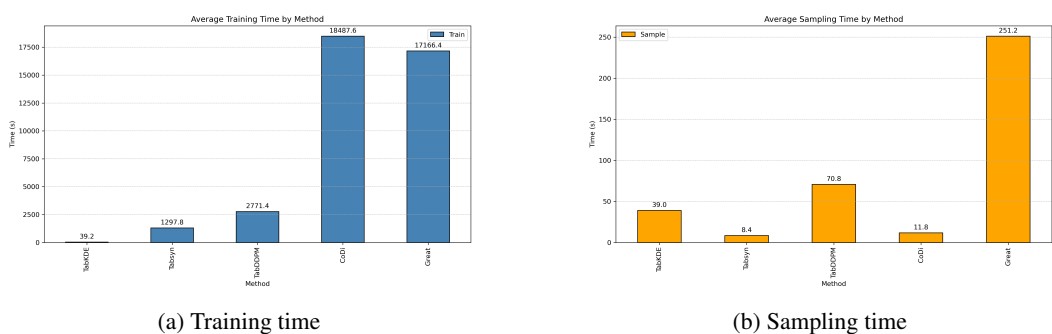

(a) Training time         (b) Sampling time

Figure 5: Average training and sampling time over Adult, Default, Shoppers, and Magic for different methods.

Table 12: Average training and sampling time for each method.

| Method | Train Time (s) | Sample Time (s) |
|---|---|---|
| Great | 17112.4 | 251.2 |
| Codi | 18487.6 | 11.8 |
| TabDDPM | 2771.4 | 70.8 |
| TABSYN | 1297.8 | 8.4 |
| TABKDE | 39.2 | 39.0 |

Table 13: Average training and sampling times (in seconds)± standard deviation for each dataset using the TABKDE model. Values are reported as mean and standard deviation over 10 repeated runs.

| Dataset | Train Time (s) | Sample Time (s) |
|---------|---------------|-----------------|
| Adult | $51.50_{\pm 1.08}$ | $50.30_{\pm 0.67}$ |
| Default | $45.00_{\pm 0.67}$ | $55.10_{\pm 0.99}$ |
| Shoppers | $21.50_{\pm 0.53}$ | $18.20_{\pm 0.42}$ |
| Magic | $33.00_{\pm 0.67}$ | $21.00_{\pm 0.67}$ |
| Beijing | $63.00_{\pm 0.47}$ | $51.80_{\pm 0.79}$ |
| News | $64.30_{\pm 1.16}$ | $166.80_{\pm 1.32}$ |
| **Average** | $\mathbf{46.38}_{\pm 0.29}$ | $\mathbf{60.53}_{\pm 0.42}$ |

# D    ACCURACY EVALUATION

In this section, we evaluate the quality of the generated synthetic data using three criteria: (1) marginal distribution alignment, (2) pairwise correlation matching, and (3) finally global alignment between synthetic and hold-out distributions is compared by how well a classifier can separate the distributions.

## D.1    MARGINAL DISTRIBUTION ALIGNMENT

When evaluating synthetic tabular data, **marginal distribution alignment score** assesses how closely each individual column matches its real-data distribution represented by train data. Following what was done in the TABSYN paper, we calculate the Kolmogorov–Smirnov (KS) distance for numerical attributes in Num and the Total Variation Distance for categorical and ordinal attributes in Cat and Ord. Table 14 presents, for each dataset, the average marginal alignment errors across all features for each method. Table 15 presents the performance of the TABKDE model in aligning marginal distributions, averaged over 10 runs.

Table 14: Performance comparison on marginal distribution alignment (Error rate %). Lower values indicate better performance. The values in parentheses denote the ratio relative to the smallest value. Baselines taken from Zhang et al. (2024)

| Method | Adult | Default | Shoppers | Magic | Beijing | News | Average |
|--------|-------|---------|----------|-------|---------|------|---------|
| SMOTE (our reproduction) | 1.63 (2.55) | 1.70 (1.49) | 2.66 (2.16) | 1.37 (1.93) | 2.10 (1.62) | 5.47 (3.18) | 2.49 (1.75) |
| CTGAN | 16.84 (26.31) | 16.83 (14.76) | 21.15 (17.20) | 9.81 (13.82) | 21.39 (16.45) | 16.09 (9.35) | 17.02 (11.99) |
| TVAE | 14.22 (22.22) | 10.17 (8.92) | 24.51 (19.93) | 8.25 (11.62) | 19.16 (14.74) | 16.62 (9.66) | 15.49 (10.91) |
| GOGGLE | 16.97 (26.52) | 17.02 (14.93) | 22.33 (18.15) | 1.90 (2.68) | 16.93 (13.02) | 25.32 (14.72) | 16.74 (11.79) |
| GReaT | 12.12 (18.94) | 19.94 (17.49) | 14.51 (11.80) | 16.16 (22.76) | 8.25 (6.35) | – | 14.20 (10.00) |
| STaSy | 11.29 (17.64) | 5.77 (5.06) | 9.37 (7.62) | 6.29 (8.86) | 6.71 (5.16) | 6.89 (4.01) | 7.72 (5.44) |
| CoDi | 21.38 (33.41) | 15.77 (13.82) | 31.84 (25.89) | 11.56 (16.28) | 16.94 (13.03) | 32.27 (18.78) | 21.63 (15.23) |
| TabDDPM | 1.75 (2.73) | 1.57 (1.42) | 2.72 (2.21) | 1.01 (1.42) | 1.30 (1.00) | 78.75 (45.83) | 14.52 (10.23) |
| TabSYN (Our reproduction) | 0.64 (1.00) | 1.14 (1.00) | 1.23 (1.00) | 0.98 (1.38) | 2.79 (2.15) | 1.72 (1.00) | 1.42 (1.00) |
| COPULADIFF | 2.01 (3.14) | 1.47 (1.29) | 2.47 (2.01) | 0.94 (1.32) | 2.13 (1.64) | 2.44 (1.42) | 1.91 (1.35) |
| VAESIMPLEKDE | 3.23 (5.05) | 7.72 (6.77) | 6.78 (5.51) | 3.12 (4.39) | 7.12 (5.48) | 10.03 (5.83) | 6.33 (4.46) |
| VAETABKDE | 3.80 (5.94) | 5.84 (5.12) | 6.31 (5.13) | 0.71 (1.0) | 4.94 (3.80) | 4.45 (2.59) | 4.34 (3.06) |
| SIMPLE-KDE | 1.92 (3.00) | 3.33 (2.92) | 3.12 (2.54) | 3.59 (5.06) | 10.32 (7.94) | 7.36 (4.28) | 4.94 (3.48) |
| RANDCORETABKDE | 1.61 (2.52) | 1.76 (1.54) | 2.54 (2.07) | 1.01 (1.42) | 1.70 (1.31) | 2.59 (1.51) | 1.87 (3.48) |
| CORETABKDE | 3.63 (5.67) | 3.29 (2.89) | 3.23 (2.63) | 1.08 (1.52) | 3.20 (2.46) | 2.87 (1.67) | 2.88 (2.03) |
| TABKDE | 1.56 (2.44) | 1.55 (1.36) | 2.44 (1.98) | 0.78 (1.1) | 1.37 (1.05) | 2.52 (1.47) | 1.70 (1.2) |

Table 15: Performance comparison on marginal distribution alignment (Error rate %) for each dataset using TABKDE model. Values are reported as mean and standard deviation over 10 repeated runs.

| Metric | Adult | Default | Shoppers | Magic | Beijing | News | Average |
|--------|-------|---------|----------|-------|---------|------|---------|
| Marginal alignment error | $1.54 \pm 0.03$ | $1.53 \pm 0.05$ | $2.46 \pm 0.09$ | $0.80 \pm 0.08$ | $1.40 \pm 0.04$ | $2.53 \pm 0.04$ | $1.71 \pm 0.05$ |

Figure 6 provides a visual comparison between some representative selected real marginal distributions and those generated by TABSYN (orange) and TABKDE (green) against the real data distributions (blue). Each row shows 4 columns from a data set. For IBM data set (bottom row)

we use COPULADIFF (orange) instead of TABSYN since it cannot scale to this large data set. It is apparent that TABKDE usually does as well and better than TABSYN. In particular, on numerical data (where continuous distributions are shown), TABKDE appears to match the real data much closer, but on categorical data, and when there are spikes in numerical data, TABKDE can have a bit more error. Since the KS distance is a worst case, it is very unforgiving for such errors on discrete data, and explains why TABKDE and TABSYN appear comparable in these marginal plots, but TABSYN has consistently smaller scores in Table 14. An average error measure on on numerical data should show an advantage for TABKDE.

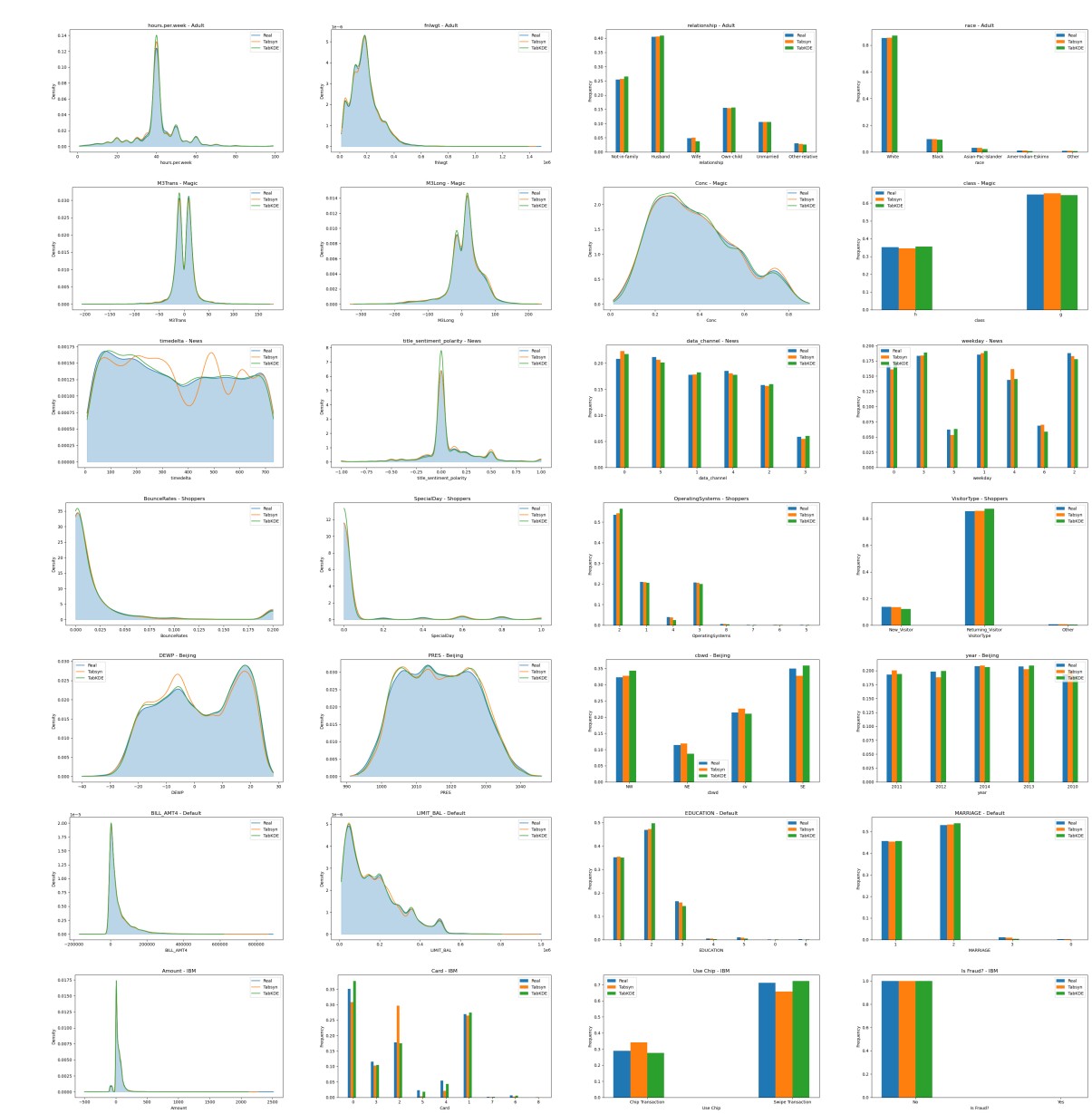

Figure 6: Marginals comparison between real data (blue), TABKDE (green), and TABSYN (orange). Each row is a data set, and sample column marginals are shown for each; some categorical and some numerical. For IBM data set (last row), TABSYN is replaced with PGE-TABSYN since TABSYN runs out of memory.

## D.2 PAIRWISE CORRELATION ALIGNMENT

We next measure pairwise correlation between columns. For numerical-numerical pairs, we can use standard Pearson correlations. However for pairs that involve categorical (and to align with measures in the TABSYN paper, we treat ordinal as categorical), we use contingency-table Total Variation Distances. In both metrics, smaller error values indicate that the synthetic table is more faithful to the original data. Table 16 presents, for each dataset, the average pairwise correlation alignment errors, computed as $(1 - \text{score})$, across all features for each method. A heatmap visualization of the divergence between the pairwise correlations in the real and synthetic data is presented in Figure 7. Table 17 presents the performance of the TABKDE model in Pairwise correlation alignment error (Error rate %), averaged over 10 runs.

We observe that TABKDE has better pairwise correlation alignment than all methods except TAB-SYN, and is comparable to SMOTE; both TABKDE and SMOTE have about 2.5 the correlation discrepancy as TABSYN. The main poor correlation for TABKDE appears in Default data set, and with 'BILL_AMT3' and 'BILL_AMT4' variables which have similar but less challenges for TABSYN, as well as GreaT and CoDi.

Table 16: Performance comparison of tabular data synthesis methods based on Pairwise correlation alignment error (Error rate %). Lower values indicate better performance. The values in parentheses denote the ratio relative to the smallest value. Baselines taken from Zhang et al. (2024)

| Method | Adult | Default | Shoppers | Magic | Beijing | News | Average |
|---|---|---|---|---|---|---|---|
| SMOTE (Our reproduction) | 4.3 (2.67) | 11.54 (5.11) | 3.68 (1.47) | 1.88 (2.29) | 3.3 (1.22) | 1.67 (1.25) | 4.39 (1.99) |
| CTGAN | 20.23 (12.57) | 26.95 (11.92) | 13.08 (5.23) | 7.0 (8.54) | 22.95 (8.47) | 5.37 (4.01) | 15.93 (7.21) |
| TVAE | 14.15 (8.79) | 19.5 (8.63) | 18.67 (7.47) | 5.82 (7.10) | 18.01 (6.65) | 6.17 (4.60) | 13.72 (6.21) |
| GOGGLE | 45.29 (28.13) | 21.94 (9.71) | 23.9 (9.56) | 9.47 (11.55) | 45.94 (16.95) | 23.19 (17.31) | 28.29 (12.8) |
| GReaT | 17.59 (10.93) | 70.02 (30.98) | 45.16 (18.06) | 10.23 (12.48) | 59.6 (21.99) | – | 40.52 (18.33) |
| STaSy | 14.51 (9.01) | 5.96 (2.64) | 8.49 (3.39) | 6.61 (8.05) | 8.0 (2.96) | 3.07 (2.29) | 7.77 (3.52) |
| CoDi | 22.49 (13.98) | 68.41 (30.27) | 17.78 (7.11) | 6.53 (7.97) | 7.07 (2.61) | 11.1 (8.28) | 22.23 (10.06) |
| TabDDPM | 3.01 (1.87) | 4.89 (2.16) | 6.61 (2.64) | 1.7 (2.07) | 2.71 (1.00) | 13.16 (9.82) | 5.34 (2.42) |
| TABSYN(Our reproduction) | 1.61 (1.00) | 2.26 (1.00) | 2.5 (1.00) | 0.82 (1.00) | 4.7 (1.73) | 1.34 (1.00) | 2.21 (1.00) |
| COPULADIFF | 4.61 (2.86) | 3.29 (1.46) | 5.3 (2.12) | 1.72 (2.10) | 4.5 (1.66) | 2.1 (1.57) | 3.59 (1.62) |
| VAESIMPLEKDE | 9.86 (6.12) | 12.88 (5.70) | 9.51 (3.80) | 3.12 (3.80) | 11.51 (4.25) | 4.05 (3.02) | 8.49 (3.84) |
| VAETABKDE | 7.23 (4.49) | 12.71 (5.62) | 9.68 (3.87) | 3.95 (4.82) | 9.87 (3.64) | 3.67 (2.74) | 7.85 (3.55) |
| SIMPLE-KDE | 4.64 (2.88) | 5.16 (2.28) | 5.26 (2.10) | 3.3 (4.02) | 4.72 (1.74) | 2.96 (2.21) | 4.34 (1.96) |
| RANDCORETABKDE | 3.93 (2.44) | 13.66 (6.04) | 4.27 (1.71) | 4.76 (5.80) | 4.05 (1.49) | 2.61 (1.95) | 5.46 (2.47) |
| CORETABKDE | 6.3 (3.91) | 9.91 (4.39) | 5.77 (2.31) | 2.18 (2.66) | 5.86 (2.16) | 2.82 (2.10) | 5.47 (2.48) |
| TABKDE | 4.51 (2.80) | 9.93 (4.40) | 4.31 (1.72) | 2.72 (3.32) | 3.74 (1.38) | 2.83 (2.11) | 4.67 (2.11) |

Table 17: Performance comparison on pairwise correlation alignment (Error rate %) for each dataset using TABKDE model. Values are reported as mean and standard deviation over 10 repeated runs.

| Metric | Adult | Default | Shoppers | Magic | Beijing | News | Average |
|---|---|---|---|---|---|---|---|
| Pairwise Corr. Align. Error | $4.05 \pm 0.27$ | $11.33 \pm 1.49$ | $4.39 \pm 0.16$ | $2.80 \pm 0.68$ | $3.80 \pm 0.22$ | $2.95 \pm 0.17$ | $4.89 \pm 0.47$ |

We also compare against some variants on the larger IBM dataset. Recall that on our laptop CPU, neither TABSYN or SMOTE can run on this data set – they both run out of memory. Instead we compare TABKDE against our baselines including COPULADIFF and PGE-TABSYN. Also, note that we apply a modeling trick with Zip / Merchant State / Merchant City and with MCC / Merchant Name with TABKDE but not PGE-TABSYN and TABSYN (on GPU). Recall also that TABKDE (about 10 minutes training time on CPU; 40 seconds on GPU) was much faster than either of COPULADIFF (about 15 hours on CPU), PGE-TABSYN (over 40 hours on CPU) or TABSYN TABSYN (OOM on CPU, about 20 minutes on GPU). The average pairwise correlation alignment error for TABSYN and PGE-TABSYN (without modeling trick) are %40.42 and %30.53, respectively, while for TABKDE and COPULADIFF (with modeling trick), are is %25.29 and %22.59; see Table 18. Indeed as shown in Figure 8 of pairwise correlation plots, the methods work largely similar except on the highly correlated pairs where we employ the modeling trick.

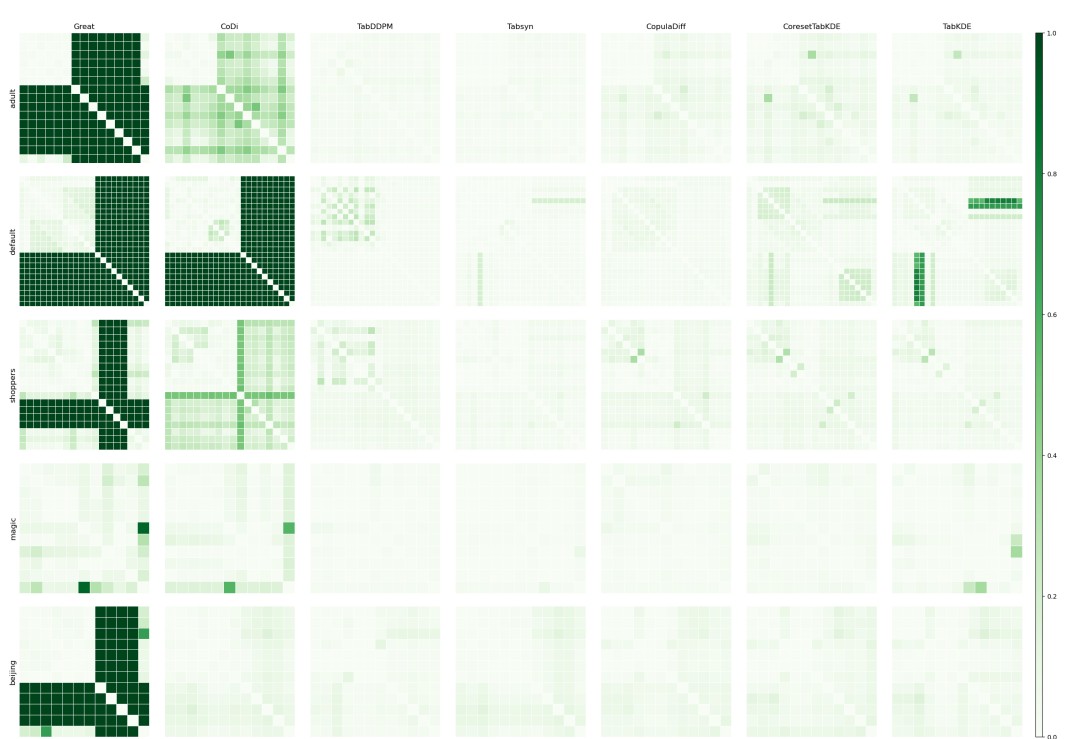

Figure 7: Pairwise correlation Divergence plots for each dataset and methods.

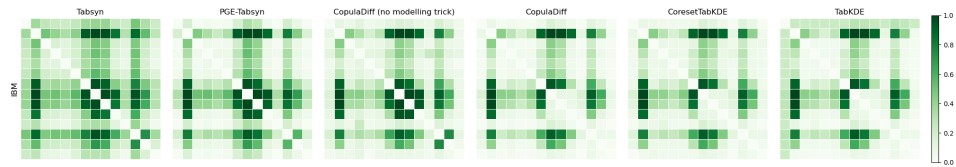

Figure 8: Pairwise correlation Divergence plots for IBM dataset and methods: TABSYN, PGE-TABSYN, COPULADIFF (with and without modeling trick), CORETABKDE, TABKDE. For CORETABKDE, the size of coreset and the bandwidth are 5,000 and .15, respectively.

Table 18: Marginal and Pairwise accuracy on IBM dataset. See the last paragraph of Subsection B.1 for reduced modeling explanation.

| Method | Marginal Alignment | Pairwise Correlation | Reduced Modeling |
|---|---|---|---|
| TABSYN | 16.99 | 40.42 | no |
| PGE-TABSYN | 9.29 | 30.53 | no |
| COPULADIFF | 6.81 | 29.42 | no |
| COPULADIFF | 3.59 | 22.59 | yes |
| CORETABKDE | 5.25 | 23.93 | yes |
| TABKDE | 3.58 | 25.29 | yes |

## D.3 GLOBAL DISTRIBUTION ALIGNMENT

High-quality synthetic data should be able to take the place of real data, letting us train models on it for downstream tasks like classification or regression and have indistinguishable performance (and without revealing private information). We assess by building a classifier to attempt to distinguish between the synthetic data and a split of data holdout from the training process. We consider two standard classifiers: XGBoost and logistic regression.

First, following standard practice, we use XGBoost to build a classifier on the synthetic data, and measure the AUC for classification tasks, and RMSE for regression tasks. This is referred to as *machine learning efficiency*. The results in Table 19 on baselines from the TABSYN paper, as well as SMOTE, and other baselines in our model. Note that TABSYN appears twice in this table (as well as in Tables 21 and 24), to include the value recorded in Zhang et al. (2024) and then our reproduction of the result. The Real row in Table 19 shows the ideal performance, using true training data to train the classifier or regression models. The Average Gap column indicates the percentage drop in performance when synthetic data is used to train the classifier or regression models. Table 20 presents the mean and standard deviation of machine learning efficiency (MLE) over 10 runs, indicating that the TABKDE model demonstrates strong robustness with respect to this metric.

Again TABSYN provides the best error, and TABKDE is nearly as good – almost within the natural variation in TABSYN's performance. Perhaps surprisingly, the simple method SMOTE also performs on par with TABSYN and TABKDE.

Table 19: Machine learning efficiency performance comparison across datasets. The results of baselines above the first line are taken from Zhang et al. (2024). To compute the Average Gap, we take the average across all datasets of the relative difference between the performance of a model trained on synthetic data ($s_i$) and the performance of the same model trained on real data ($r_i$); Average Gap $= \frac{1}{N} \sum_{i=1}^{N} \left( \frac{|s_i - r_i|}{r_i} \right) \times 100$.

| Methods | Adult (AUC↑) | Default (AUC↑) | Shoppers (AUC↑) | Magic (AUC↑) | Beijing (RMSE↓) | News (RMSE↓) | Average Gap (%) |
|---|---|---|---|---|---|---|---|
| Real | 0.927 | 0.770 | 0.926 | 0.946 | 0.423 | 0.842 | 0% |
| SMOTE | 0.899 | 0.741 | 0.911 | 0.934 | 0.593 | 0.897 | 9.39% |
| CTGAN | 0.886 | 0.696 | 0.875 | 0.855 | 0.902 | 0.880 | 24.5% |
| TVAE | 0.878 | 0.724 | 0.871 | 0.887 | .770 | 1.01 | 20.9% |
| GOGGLE | 0.778 | 0.584 | 0.658 | 0.654 | 1.09 | 0.877 | 43.6% |
| GReaT | 0.913 | 0.755 | 0.902 | 0.888 | 0.653 | OOM | 13.3% |
| STaSy | 0.906 | 0.752 | 0.914 | 0.934 | 0.656 | 0.871 | 10.9% |
| CoDi | 0.871 | 0.525 | 0.865 | 0.932 | 0.818 | 1.21 | 30.5% |
| TabDDPM | 0.907 | 0.758 | 0.918 | 0.935 | 0.592 | 4.86 | 9.14% |
| TABSYN(Our reproduction) | 0.911 | 0.760 | 0.913 | 0.942 | 0.663 | 0.820 | 10.70% |
| COPULADIFF | 0.901 | 0.763 | 0.912 | 0.939 | 0.667 | 0.921 | 12.18% |
| VAESIMPLEKDE | 0.896 | 0.733 | 0.874 | 0.912 | 0.777 | 1.037 | 20.70% |
| VAETABKDE | 0.890 | 0.747 | 0.871 | 0.913 | 0.649 | 0.860 | 11.20% |
| SIMPLE-KDE | 0.901 | 0.730 | 0.913 | 0.931 | 0.756 | 1.167 | 21.39% |
| RANDCORETABKDE | 0.883 | 0.730 | 0.911 | 0.929 | 0.713 | 0.881 | 14.42% |
| CORETABKDE | 0.881 | 0.712 | 0.919 | 0.928 | 0.744 | 0.877 | 15.86% |
| TABKDE | 0.906 | 0.745 | 0.917 | 0.934 | 0.675 | 0.869 | 11.76% |

Second, we use a logistic regression classifier. Follow standard practice we now use this to try to separate the synthetic data from the heldout data. We quantify this as a classifier two-sample test (C2ST) as provided by SDMetrics; larger values closer to 1 are better. Table 21 shows both a

Table 20: Machine learning efficiency comparison across datasets for the TABKDE model, reporting accuracy as mean $\pm$ standard deviation over 10 runs.

|  | Adult | Default | Shoppers | Magic | Beijing | News |
|---|---|---|---|---|---|---|
| MLE | $0.904 \pm 0.003$ | $0.744 \pm 0.012$ | $0.916 \pm 0.006$ | $0.931 \pm 0.004$ | $0.678 \pm 0.010$ | $0.852 \pm 0.021$ |

comparison drawn directly from the TABSYN paper against a variety of recent baselines; below the line we reproduce results on TABSYN, show results for SMOTE, and variants of our method TABKDE.

As before, TABKDE is roughly the same as SMOTE with $0.93$ and only bested by TABSYN which has about $0.97$. Other baselines achieve $0.79$ (TabDDPM) or below $0.66$.

Table 21: C2ST Scores of generative models on tabular datasets. The results of baselines above the first line are taken from Zhang et al. (2024)

| Method | Adult | Default | Shoppers | Magic | Beijing | News | Average |
|---|---|---|---|---|---|---|---|
| CTGAN | 0.5949 | 0.4875 | 0.7488 | 0.6728 | 0.7531 | 0.6947 | 0.6586 |
| TVAE | 0.6315 | 0.6547 | 0.2962 | 0.7706 | 0.8659 | 0.4076 | 0.6044 |
| GOGGLE | 0.1114 | 0.5163 | 0.1418 | 0.3262 | 0.4779 | 0.0745 | 0.2747 |
| GReaT | 0.5376 | 0.4710 | 0.4285 | 0.4326 | 0.6893 | — | 0.5118 |
| STaSy | 0.4054 | 0.6814 | 0.5482 | 0.6939 | 0.7922 | 0.5287 | 0.6083 |
| CoDi | 0.2077 | 0.4595 | 0.2784 | 0.7206 | 0.7177 | 0.0201 | 0.4007 |
| TabDDPM | 0.9755 | 0.9712 | 0.8349 | 0.9998 | 0.9513 | 0.0002 | 0.7888 |
| SMOTE | 0.9212 | 0.9332 | 0.9107 | 0.9803 | 0.9972 | 0.8633 | 0.9334 |
| TABSYN(Our reproduction) | 0.9949 | 0.9804 | 0.9699 | 0.9893 | 0.9268 | 0.9584 | 0.9699 |
| COPULADIFF | 0.8557 | 0.9798 | 0.8665 | 0.9914 | 0.9576 | 0.9793 | 0.9384 |
| VAESIMPLEKDE | 0.7199 | 0.4082 | 0.6736 | 0.9665 | 0.7392 | 0.3782 | 0.6476 |
| VAETABKDE | 0.7483 | 0.4828 | 0.7242 | 0.9984 | 0.8022 | 0.8075 | 0.7606 |
| SIMPLE-KDE | 0.9196 | 0.8716 | 0.8110 | 0.9711 | 0.9497 | 0.4975 | 0.8368 |
| RANDCORETABKDE | 0.9215 | 0.9570 | 0.8757 | 0.9921 | 0.9503 | 0.8901 | 0.9311 |
| CORETABKDE | 0.8254 | 0.873 | 0.8462 | 0.9864 | 0.8924 | 0.8643 | 0.8813 |
| TABKDE | 0.9219 | 0.9579 | 0.9161 | 1.0000 | 0.9514 | 0.8819 | 0.9382 |

## D.4 Precision and Recall

$\alpha$-Precision and $\beta$-Recall are two complementary metrics used to assess the quality of synthetic tabular data, as used in the TABSYN paper (Zhang et al., 2024). $\alpha$-Precision measures the fidelity of the synthetic data to the real data, indicating how well the synthetic samples preserve fine-grained details and local structures. A higher $\alpha$-Precision score reflects greater similarity to the original data. In contrast, $\beta$-Recall evaluates the extent to which the synthetic data covers the real data distribution, with higher scores indicating broader and more diverse coverage of the feature space. An ideal generative model should balance both metrics—achieving high $\alpha$-Precision while also maintaining strong $\beta$-Recall—thus producing synthetic data that is both accurate and representative of the true distribution. Tables 22 and 23 summarize $\alpha$-Precision and $\beta$-Recall scores.

TABSYN does the best on $\alpha$-precision, but SMOTE does better on $\beta$-recall. On $\alpha$-precision TABKDE (95%) nearly matches TABSYN (98.7%), and is better than any other method (including SMOTE), except our variant COPULADIFF which reaches (96%). On $\beta$-precision, TABKDE (42%) almost matches TABSYN (48%) as is almost as good as any other method with GReaT and STaSy slightly better (43%); other than SMOTE (78%). But as we discuss next, this likely because SMOTE generates data mirroring some of the training data.

Table 22: $\alpha$-Precision scores for various methods on the 6 standard data sets. The last column shows the average score and rank. Scores above the line are reproduced from Zhang et al. (2024) with their data split; below the line are with equal-sized split.

| Method | Adult | Default | Shoppers | Magic | Beijing | News | Average | Ranking |
|---|---|---|---|---|---|---|---|---|
| CTGAN | 77.74 | 62.08 | 76.97 | 86.90 | 96.27 | 96.96 | 82.82 | 12 |
| TVAE | 98.17 | 85.57 | 58.19 | 86.19 | 97.20 | 86.41 | 85.29 | 10 |
| GOGGLE | 50.68 | 68.89 | 86.95 | 90.88 | 88.81 | 86.41 | 78.77 | 15 |
| GReaT | 55.79 | 85.90 | 78.88 | 85.46 | 98.32 | - | 80.87 | 13 |
| STaSy | 82.87 | 90.48 | 89.65 | 86.56 | 89.16 | 94.76 | 88.91 | 8 |
| CoDi | 77.58 | 82.38 | 94.95 | 85.01 | 98.13 | 87.15 | 87.03 | 9 |
| TabDDPM | 96.36 | 97.59 | 88.55 | 98.59 | 97.93 | 0.00 | 79.83 | 14 |
| TABSYN | 99.52 | 99.26 | 99.16 | 99.38 | 98.47 | 96.80 | 98.67 | 1 |
| smote | 92.83 | 98.40 | 92.60 | 96.76 | 98.64 | 87.93 | 94.52 | 6 |
| COPULADIFF | 98.09 | 98.99 | 95.43 | 98.43 | 97.33 | 93.98 | 97.04 | 2 |
| VAESIMPLEKDE | 88.21 | 80.90 | 82.46 | 7.03 | 75.56 | 19.29 | 72.24 | 16 |
| VAETABKDE | 98.39 | 91.71 | 97.36 | 98.50 | 93.29 | 84.86 | 94.02 | 7 |
| SIMPLE-KDE | 98.10 | 93.88 | 98.84 | 90.13 | 96.41 | 29.25 | 84.44 | 11 |
| RANDCORETABKDE | 95.67 | 4.62 | 91.64 | 98.68 | 98.11 | 96.68 | 95.90 | 3 |
| CORETABKDE | 98.01 | 89.44 | 90.27 | 99.12 | 95.70 | 94.96 | 94.58 | 5 |
| TABKDE | 94.46 | 94.45 | 92.18 | 98.98 | 97.47 | 97.48 | 95.83 | 4 |

Table 23: $\beta$-Recall scores for various methods on the 6 standard data sets. The last column shows the average score and rank. Scores above the line are reproduced from Zhang et al. (2024) with their data split; below the line are with equal-sized split.

| Method | Adult | Default | Shoppers | Magic | Beijing | News | Average | Ranking |
|---|---|---|---|---|---|---|---|---|
| CTGAN | 30.80 | 18.22 | 31.80 | 11.75 | 34.80 | 24.97 | 25.39 | 16 |
| TVAE | 38.87 | 23.13 | 19.78 | 32.44 | 28.45 | 29.66 | 28.72 | 15 |
| GOGGLE | 8.80 | 14.38 | 9.79 | 9.88 | 19.87 | 2.03 | 10.79 | 17 |
| GReaT | 49.12 | 42.04 | 44.90 | 34.91 | 43.34 | OOM | 42.86 | 6 |
| STaSy | 29.21 | 39.31 | 37.24 | 53.97 | 54.79 | 39.42 | 42.99 | 5 |
| CoDi | 9.20 | 19.94 | 20.82 | 50.56 | 52.19 | 34.40 | 31.19 | 13 |
| TabDDPM | 47.05 | 47.83 | 47.79 | 48.46 | 56.92 | 0.00 | 41.34 | 7 |
| TABSYN | 47.56 | 48.00 | 48.95 | 48.03 | 55.48 | 45.04 | 48.84 | 2 |
| SMOTE | 76.88 | 76.00 | 77.09 | 82.45 | 79.22 | 80.00 | 78.60 | 1 |
| COPULADIFF | 41,29 | 46.21 | 43.21 | 46.38 | 51.65 | 43.86 | 45.43 | 3 |
| VAESIMPLEKDE | 38.78 | 20.71 | 38.01 | 40.13 | 45.62 | 2.00 | 30.87 | 14 |
| VAETABKDE | 43.68 | 27.32 | 48.81 | 45.56 | 51.12 | 12.64 | 38.18 | 9 |
| SIMPLE-KDE | 45.90 | 36.60 | 43.12 | 44.78 | 52.68 | 3.90 | 37.83 | 11 |
| RANDCORETABKDE | 37.67 | 35.81 | 44.77 | 44.82 | 51.56 | 17.54 | 38.69 | 8 |
| CORETABKDE | 27.46 | 20.86 | 39.40 | 40.35 | 48.30 | 13.01 | 31.56 | 12 |
| TABKDE | 48.54 | 43.05 | 47.22 | 48.80 | 54.39 | 17.82 | 43.30 | 4 |

## E PRIVACY PRESERVATION

Finally, we evaluate how well we can preserve the privacy of the training data in the synthetic data generation process. We use the distance to closest record (DCR) function in the latent space to evaluate this. That is for each synthetic data point generated, we both look at the distribution of distances to training or held-out data, and also whether the closest record was from the held-out or training set. An ideal synthetic distribution would match the distance distribution of the training data to the heldout data, and would be roughly equally likely to be close to the heldout and training data.

First Table 24 we calculate the "DCR score" which is the percentage of synthetic data closer to training data than held-out; ideally we would like this to be close to 50%. This was proposed by the recent TABDIFF paper (Shi et al., 2025), and we reproduce their results in the top half of the table, and show our methods (and SMOTE and TABSYN) below the line on an equal split. We see most diffusion methods (including our COPULADIFF) can consistently achieve below 52%. Our main method TABKDE obtains an average DCR score of about 58%, which is servicable.

On the other hand, SMOTE has an average DCR score of 95%. This indicates that it often nearly replicates the training data. Its method chooses a training record, finds the $k$ nearest neighbor, and selects a new point in the convex combination of these points, then de-tokenizes back to the tabular format. Because it works with a one-hot encoding, probably most records map back to the same discrete values as the first record, and it often failures to generate substantially new data, hence leaking the training data.

Table 24: The DCR score indicates the likelihood that a generated data sample resembles the training set more than the test set. A value nearer to 50% is considered ideal. Values above the line reproduced from Shi et al. (2025) with their train-held-out split.

| Method | Adult | Default | Shoppers | Beijing | News | Average |
|---|---|---|---|---|---|---|
| STaSy | 50.33 (1.01) | 50.23 (1.00) | 51.53 (1.03) | 50.59 (1.01) | 50.59 (1.00) | 50.65 (1.00) |
| CoDi | 49.92 (1.00) | 51.82 (1.03) | 51.06 (1.02) | 50.87 (1.01) | 50.79 (1.00) | 50.89 (1.01) |
| TabDDPM | 51.14 (1.02) | 52.15 (1.04) | 63.23 (1.26) | 80.11 (1.59) | 79.31 (1.57) | 65.19 (1.29) |
| TABSYN | 50.94 (1.02) | 51.20 (1.02) | 52.90 (1.06) | 50.37 (1.00) | 50.85 (1.01) | 51.65 (1.02) |
| TABDIFF | 50.10 (1.00) | 51.11 (1.02) | 50.24 (1.00) | 50.50 (1.00) | 51.04 (1.01) | 50.60 (1.00) |
| COPULADIFF | 50.34 (1.01) | 50.96 (1.01) | 50.72 (1.01) | 50.29 (1.00) | 53.00 (1.05) | 51.06 (1.01) |
| VAESIMPLEKDE | 61.33 (1.23) | 58.08 (1.16) | 58.83 (1.17) | 60.73 (1.21) | 59.00 (1.17) | 59.59 (1.18) |
| VAETABKDE | 61.28 (1.23) | 57.87 (1.15) | 57.70 (1.15) | 60.42 (1.20) | 58.00 (1.15) | 59.45 (1.17) |
| smote | 91.18 (1.83) | 91.46 (1.82) | 96.76 (1.92) | 100.00 (1.99) | 99.00 (1.96) | 95.68 (1.89) |
| SIMPLE-KDE | 63.32 (1.27) | 63.49 (1.26) | 58.18 (1.16) | 55.42 (1.10) | 56.12 (1.11) | 59.71 (1.18) |
| TABSYN | 51.33 (1.03) | 51.61 (1.03) | 51.99 (1.03) | 53.20 (1.06) | 51.00 (1.01) | 51.83 (1.02) |
| RANDCORETABKDE | 62.30 (1.24) | 63.09 (1.26) | 58.91 (1.17) | 63.50 (1.26) | 55.59 (1.10) | 60.68 (1.20) |
| CORETABKDE | 52.59 (1.05) | 54.11 (1.08) | 55.04 (1.1) | 51.17 (1.03) | 52.00 (1.03) | 52.98 (1.05) |
| TABKDE | 62.23 (1.25) | 63.46 (1.26) | 58.80 (1.17) | 54.24 (1.11) | 54.54 (1.08) | 58.55 (1.16) |

If the closest record is from the training or heldout data is an imperfect measure of privacy, since there may be heldout data nearly as close. One way to evaluate this is to consider the distribution of how close the synthetic data to the heldout (red) matches the distribution of the synthetic data to the train (blue). We show this in Figure 9 for TABKDE, TABSYN, and SMOTE for the 6 standard datasets. We observe that for TABKDE and TABSYN these distributions are multi-modal, but still match almost perfectly for each data set. On the other hand SMOTE has a very different distribution, and the synthetic to train (blue) is always much smaller (almost always close to 0 for Adult, Default, Shopping, and Beijing), indicating it is too closely just reproducing the training data.

## F CORESETS FOR GENERATIVE TABULAR DATA MODELING

A *coreset* (Phillips, 2016) is a compact, weighted set of points that provides a close approximation to the full dataset for a specific downstream task. In the context of Kernel Density Estimation (KDE), a coreset serves to approximate the full KDE using significantly fewer, yet strategically chosen, representative points. A *weak coreset* is a coreset that the set is not necessarily a subset of the original point set. In this setting, the "weak" aspect will turn out to be strategically advantageous.

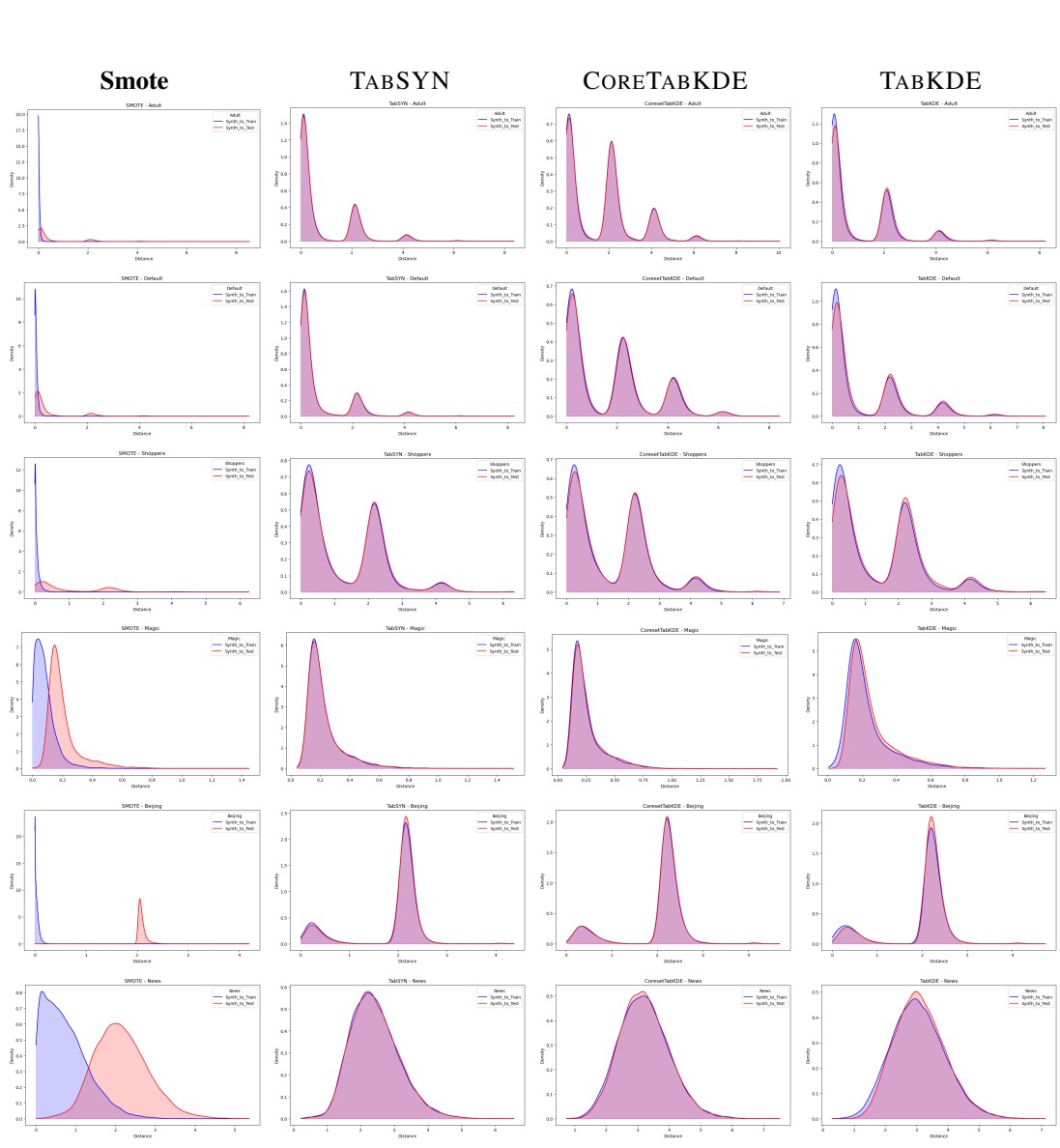

Figure 9: Privacy comparison based on DCR distributions for synthetic to training data (blue) and synthetic to held-out data (red). Each row is a data set, the columns show results for SMOTE, TABSYN, CORETABKDE, and TABKDE.

Our proposed TABKDE framework employs the full KDE to generate samples from the Copula latent representation $Z \subset [0,1]^{n \times d}$ of the tabular data $\mathcal{T}$. The full KDE over $Z$ is given by:

$$f_Z(z) = \frac{1}{|Z|} \sum_{z_i \in Z} K\left(\frac{z - z_i}{h}\right),$$

which we here consider it as the ground-truth likelihood function over $z \in [0,1]^d$. To approximate $f_Z(\cdot)$ using a weak coreset, we define a parameterized KDE $\tilde{f}_\Theta(\cdot)$ based on a small set of learnable coreset (support) points $Q = \{q_1, \ldots, q_m\}$ and their corresponding non-negative weights $W = \{\omega_1, \ldots, \omega_m\}$, constrained such that $\sum_{i=1}^m \omega_i = 1$, where $\Theta = \{Q, W\}$. The approximated density function is:

$$\tilde{f}_\Theta(z) = \sum_{i=1}^m \omega_i K\left(\frac{z - q_i}{h}\right),$$

with $m \ll n$. It is known (Phillips and Tai, 2020) that a sample $Q \sim Z$ of $m = O((1/\varepsilon^2)\log(1/\delta))$ points already ensures a strong $L_\infty$ coresets approximation that $\|f_Z - f_Q\|_\infty \leq \varepsilon$ with probability at least $1 - \delta$. Note that for a fixed kernel $K$, this bound is independent of the dimension $d$. While we use this as a starting point, we seek to improve it with a weak coreset.

In particular, the parameters $\Theta$ are optimized by minimizing the expected squared $L_2$ deviation between the full KDE and its coreset approximation, evaluated over samples drawn from the uniform distribution on $[0,1]^d$:

$$\mathbb{E}_{z \sim \mathsf{Unif}([0,1]^d)}\left[\left(\tilde{f}_\Theta(z) - f_Z(z)\right)^2\right].$$

By optimizing the positions and weights of the coreset points to minimize the discrepancy between the coreset-based KDE and the full KDE, the method preserves key distributional features, such as modes, spread, and overall shape. Moreover, because we use a weak coreset, we are not replicating the training data, and this minimizes the risk of overfitting to the data or leaking its private attributes.

This objective can be optimized via stochastic gradient descent (SGD) with TRAINCORESETKDE (Algorithm 14).

---

**Algorithm 14** TRAINCORESETKDE$(Z, T)$

---

1: Define $f_Z(z) = \frac{1}{|Z|} \sum_{z_i \in Z} K\left(\frac{z - z_i}{h}\right)$
2: Initialize randomly the coreset points $Q = \{q_1, \ldots, q_m\} \subset Z$
3: Initialize the weights $W = \{\omega_1, \ldots, \omega_m\}$ with $\omega_i = 1/m$
4: Define $\tilde{f}_\Theta(z) = \sum_{i=1}^m \omega_i K\left(\frac{z - q_i}{h}\right)$
5: **For** $t = 1$ to $T$:
6:     Sample $z \sim \mathsf{Unif}([0,1]^d)$
7:     Compute loss $\mathcal{L}(z) = \left(\tilde{f}_\Theta(z) - f(z)\right)^2$
8:     Update $\Theta$ via gradient descent to minimize $\mathcal{L}(z)$
9: **return** $\Theta = \{Q, W\}$

---

We employ the Gaussian kernel as $K(v) = \exp(-v^2)$ in our formulation. Once trained, the learned weak coreset $\{(q_i, \omega_i)\}_{i=1}^m$ replaces the full KDE sampling step in Algorithm 11. This modification constitutes the only difference between TABKDE and CORESETTABKDE, and is detailed in the sampling procedure below.

---

**Algorithm 15** SAMPLECORESETKDE-ITERATIVE($Q, W$)

---

1: $Q, W = $ TRAINCORESETKDE($Z, T$)
2: Sample $q_i \in Q$ with probability $\omega_i \in W$
3: Sample radius $r > 0$ from $f$
4: Sample $v \sim \mathcal{N}(0, \Sigma)$, set $u = \frac{v}{\|v\|}$
5: $z' \leftarrow z_i + r \cdot u$
6: **While** $\{j : z'_j \notin [0, 1]\} \neq \varnothing$:
7:      $J \leftarrow \{j : z'_j \notin [0, 1]\}$
8:      Sample $v' \sim \mathcal{N}(0, \Sigma)$, set $w = \frac{v'}{\|v'\|}$
9:      $s \leftarrow \frac{\|(u_k)_{k \in J}\|}{\|(w_k)_{k \in J}\|}$
10:     $u_j \leftarrow s \cdot w_j$ for each $j \in J$
11:     $z' \leftarrow z_i + r \cdot u$
12: **return** $z'$

---

We may also define a RANDOMCORESETTABKDE variant, in which the optimization step in Alg. 14 (Step 1) is omitted. Instead, a subset $Q \subset Z$ of size $m$ is sampled uniformly at random; then we simply invoke SAMPLEKDE-ITERATIVE(Q) (Alg 11) with $Q$ instead of $Z$. As noted above, this simple approach of taking a random sample $Q \sim Z$ has strong $L_\infty$ approximation guarantees on how well it approximates the KDE of $Z$; and this does not depend on either the dimension $d$ or the size $n$ of $Z$ – it only depends on the size $|Q| = m$ of the sample.

### F.1 EMPIRICAL EVALUATION OF CORESET METHODS

In this section, we empirically examine the advantages and limitations of the coreset approaches CORETABKDE and RANDCORETABKDE introduced in Subsection F. Across all the data sets, we set $m = 5,000$; as shown in Figure 10, the marginal and pairwise correlation alignment scores look to plateau around that value. By ablation study, we set $h = 0.2$ for data sets Adult, Default, Shoppers, Magic, Beijing, and News, we set $h = 0.2, 0.4, 0.2, 0.2, 0.2, 0.5$, respectively. We consider bandwidths $h \in \{0.1, 0.2, \ldots, 1.0\}$ at $0.1$ intervals, and examine the loss function of the TRAINCORESETKDE procedure run for $T = 30$ epochs. We select the bandwidth $h$ where the corresponding loss has the steepest descent towards zero. High values of $h$ result in negligible loss updates, while overly small values lead to premature convergence at suboptimal loss values. If multiple consecutive bandwidths yield similar behavior, we take the smaller as the chosen value.

While the bandwidth selection is chosen based only on the loss function in the training data, we also validate our selection on the test data. As shown on the Adult data set with $m = 5000$ in Figure 11, the alignment error (both marginal and pairwise correlation; higher better) is fairly stable across choices of $h$ in $0.1$ to $1.0$, but has a local peak around $h = 0.2$. Moreover, the DCR score has a more noticeable drop for $h \leq 0.6$; hence $h = 0.2$ is confirmed as a good choice. While this KDE is build in $[0, 1]^d$ copula space for all data sets, the dimension changes from 6 (for Adult) to 46 (for News), and the higher dimensional setting has more room to spread points out, and prefers a larger bandwidth.

The accuracy scores are reported in the numerous tables above, and one can observe this method achieves accuracy nearly as good as TABKDE, but often a bit worse; see for instance Tables 14 and 16. The scalability of RANDCORETABKDE is also about the same on the data sets we consider as TABKDE (which is already very efficient). However, now we require much less space to store the model; and the based on KDE-coreset results (Phillips and Tai, 2020), the accuracy for a fixed size coreset should not decrease as the training data grows. However, the CORETABKDE requires some optimization training time: approximately 55 minutes (on the laptop CPU) average across the five datasets: Adult, Default, Shoppers, Magic, and Beijing.

What is more interesting is the effect on privacy in using CORETABKDE. As shown in Table 24, CORETABKDE offers a notable improvement in privacy (under the unstable DCR score), with an average DCR score of about 53%. This is because the coreset no longer precisely stores the training data; rather it is storing a distribution of data points $Q$ which have a similar KDE as does the training data $Z$. Hence, when it generates synthetic data, it is not using some training data point $z \in Z$ as a starting point.

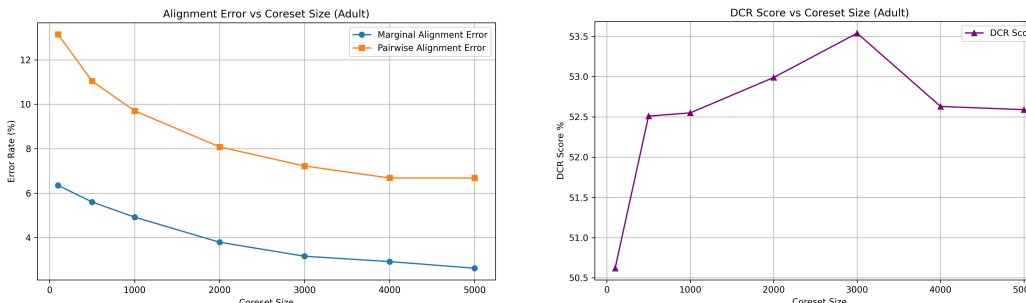

Figure 10: Left: Marginal distribution alignment error (%) and pairwise correlation alignment error (%) as coreset size increases. Right: DCR score as coreset size increases.

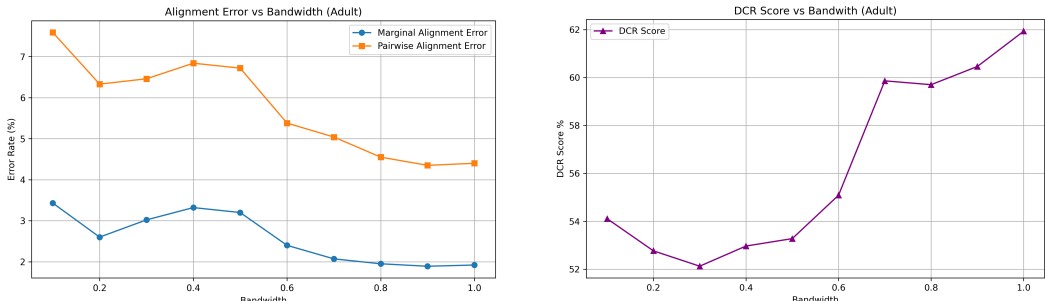

Figure 11: Left: Marginal distribution alignment error (%) and pairwise correlation alignment error (%) as bandwidth increases. Right: DCR score as bandwidth increases.

## G   ABLATION STUDY ON VARIANTS OF TABKDE

In this section we do a small ablation study to investigate the differences among variants of TABKDE. Most data is presented in earlier tables in the main paper. The main take-a-ways are as follows.

The methods that use Diffusion or VAE are significantly slower, and less scalable. As discussed above some of this can be ameliorated by avoiding one-hot encoding, but still the difference in runtime is very large on the IBM data set. Second, none of the methods that take elements from TABSYN directly match it in terms of accuracy, although COPULADIFF sometimes does nearly as well, and on average, they all also do not outperform TABKDE. Third, on privacy, COPULADIFF achieves an average DCR score of 51%, so it does quite well, improving upon TABKDE and CORETABKDE.

We next provide analysis comparing to SimpleKDE. It should already have been aparent from the accuracy evaluation where it performs a bit worse than TABKDE that it is not the preferred method. But next we provide a more in depth discussion on the marginals, where the issue becomes even more clear why the iterative element is required.

### G.1   MOTIVATION FOR TABKDE: BOUNDARY CONTROL CHALLENGES IN SIMPLEKDE

In our initial exploration, we employed **SimpleKDE** to generate synthetic samples by perturbing numerical latent representation of the data points using a Kernel Density Estimation (KDE) model. While this approach is intuitive and straightforward, it presents a critical limitation—*lack of boundary control* (see Figure 12). The perturbed samples, generated by adding Gaussian noise to real points, often fall outside the convex hull of the original dataset. This results in synthetic records that do not reflect the valid range or domain constraints of the original data, leading to unrealistic samples that may violate the natural boundaries of the feature space.

To address this issue, we developed TABKDE to better leverage the copula-transformed latent space, where all features reside. We ensure effective boundary control during sampling, maintaining the generated samples within valid limits. By resolving the boundary control problem, TabKDE produces

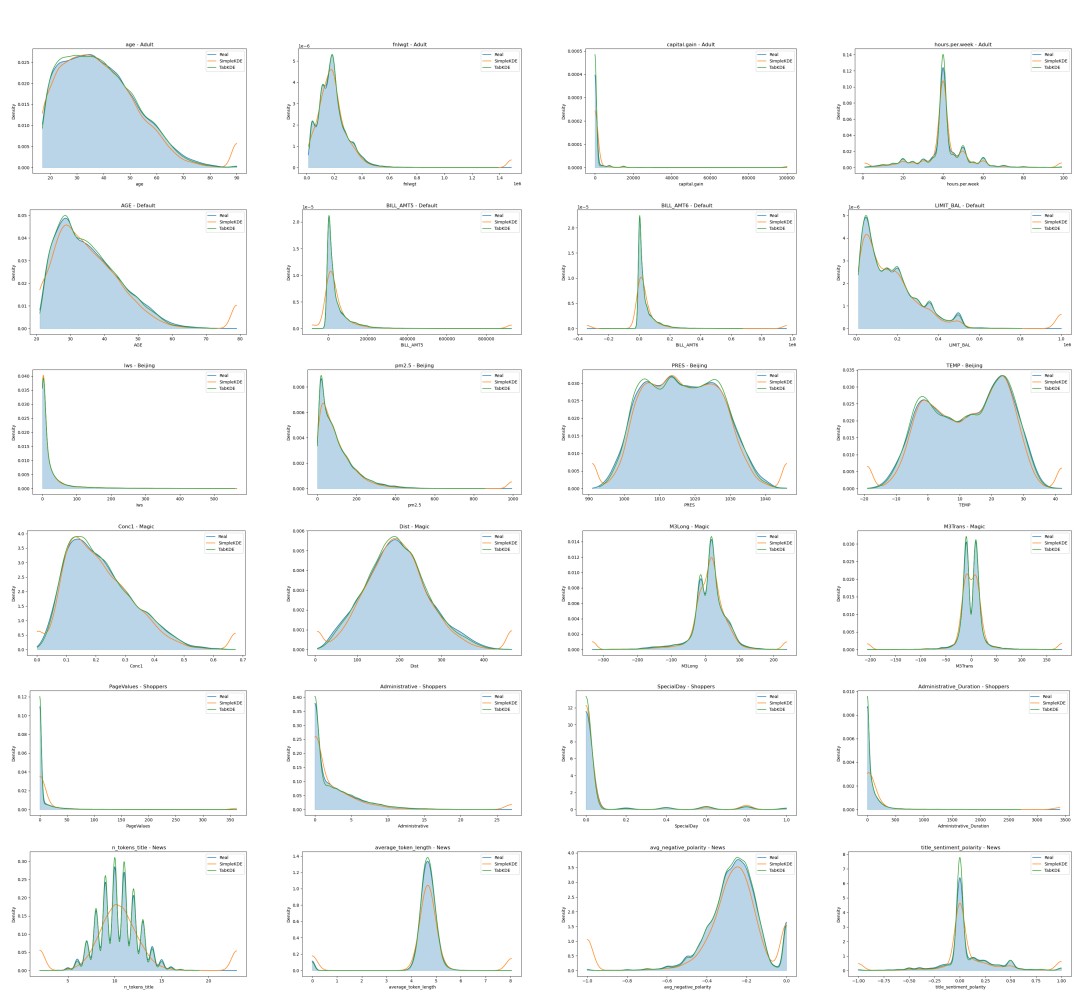

Figure 12: Marginals comparison: Simple KDE vs TABKDE

synthetic data that more accurately preserves the statistical structure and integrity of the original dataset (see Figure 12).

## H   COMPARISON WITH COPULA/KDE BASED GENERATOR MODELS

Copula-based models represent one of the earliest and most widely used approaches for synthetic tabular data generation. These methods typically rely on two steps: (i) learning marginal distributions for each feature, and (ii) coupling them via a copula function to capture dependencies across variables. Classic examples include the *GaussianCopulaSynthesizer* Patki et al. (2016), which estimates univariate marginals and applies a Gaussian copula to model correlations, and its variants such as *CopulaGAN* and vine-copula GANs, which map the copula-transformed data into a latent space before adversarial training Xu et al. (2019). While conceptually elegant, these methods often struggle with mixed-type tabular data. Categorical variables are usually handled by simplistic encodings (e.g., one-hot encoding or UniformEncoder), which can distort dependencies, and the Gaussian copula assumption may fail to capture higher-order interactions.

Recent research has extended this line of work. For example, Meyer et al. (2021) applied vine and Gaussian copulas to continuous weather data, while Majdara and Nooshabadi (2020) integrated copula transforms with diffusion-based KDE for continuous density estimation. However, these methods are limited to continuous domains and do not address challenges of mixed categorical-numerical data or scalability. Other work, such as differentially private copula models (e.g., Gambs et al. (2021)), explicitly aims to strengthen privacy at the expense of accuracy and efficiency.

Against this backdrop, TABKDE can be viewed as both building on and diverging from the copula tradition. Like classical copula generators, TabKDE maps data into a copula space (not the same as the classical copula method), standardizing marginals into the unit hypercube. However, rather than imposing parametric assumptions (e.g., Gaussian copula) or adversarial training, it employs a $d$-dimensional kernel density estimator directly in copula space. This design introduces several key innovations absent from prior copula models:

- **Principal-Guided Encoding (PGE)** for categorical features, enabling faithful one-dimensional embeddings without using one-hot encodings, which does not increase the dimension and avoids sparsity.

- **Covariance-aware geometry and boundary-respecting kernels**, which allow KDE sampling to preserve higher-order correlations and respect marginal supports.

- **DCR-calibrated kernels**, which explicitly align synthetic samples to the empirical distance-to-closest-record distribution, thereby privacy protection.

- **Coreset compression**, which produces compact, scalable generative models, in contrast to copula baselines that typically scale linearly with dataset size.

Empirical comparisons reinforce these differences. Across UCI benchmarks (Adult, Default, Magic,) and a large IBM fraud dataset, TabKDE consistently achieves lower marginal and pairwise correlation errors and substantially higher C2ST fidelity than GaussianCopulaSynthesizer, while also surpassing CopulaGAN in distributional accuracy. GaussianCopulaSynthesizer maintains slightly stronger privacy under the DCR metric (hovering near the ideal $50\%$), but this advantage stems largely from its poorer fidelity. In practice, TabKDE achieves a more balanced tradeoff: reasonable privacy coupled with diffusion-level accuracy, while retaining orders-of-magnitude efficiency gains, training in minutes compared to hours for deep copula-hybrids like CopulaGAN.

To further contextualize our approach, we compare TabKDE against both the classical *GaussianCopulaSynthesizer* and *CopulaGANSynthesizer*, in addition to our COPULADIFF variant. Table 25 reports averages across Adult, Default, Shoppers, Beijing, and News, while Tables 26 provide per-dataset breakdowns for Adult, Default, Magic, and Beijing.

**Accuracy.**   TabKDE achieves 7–10$\times$ lower distribution-alignment errors and quadruples C2ST fidelity compared to the GaussianCopulaSynthesizer. This indicates that our encoding of the copula transform together with the KDE estimator drives the observed accuracy gains.

| Metric (↓ better except C2ST ↑) | TabKDE | CopulaDiff | GaussianCopula |
|---|---|---|---|
| Average Marginal error (%) | **1.70** | 1.91 | 14.9 |
| Average Pairwise-corr error (%) | 4.67 | **3.59** | 13.3 |
| Average C2ST (↑) | **0.94** | **0.94** | 0.22 |
| DCR (ideal ≈ 50%) | 58.6 | 51.06 | **49.8** |
| Train / Sample time (s, CPU) | 39.2 / 39.0 | – | **0.9 / 6.8** |

Table 25: Comparison of TABKDE and COPULADIFF against the GaussianCopulaSynthesizer. Averages are computed over Adult, Default, Shoppers, Beijing, and News datasets.

**Speed.** GaussianCopulaSynthesizer is milliseconds-fast on CPU, effectively functioning as a restricted subset of TabKDE. Nonetheless, TabKDE remains far more efficient than deep generative models.

**Privacy.** GaussianCopulaSynthesizer yields DCR scores near the ideal $50\%$, reflecting strong privacy but poor fidelity. TabKDE (58%) attains a more balanced trade-off, offering reasonable privacy while maintaining high fidelity. Note that DCR is an imperfect metric: it only evaluates nearest-neighbor distances, and broader distributional comparisons are more favorable for TabKDE.

Overall, TabKDE consistently outperforms both complex copula-based generators (e.g., CopulaGAN) and the simpler GaussianCopula model in terms of fidelity, while remaining efficient and maintaining reasonable privacy.

### H.1 EXTENDED COPULA+KDE BASELINES

We additionally benchmark TabKDE against several methods at the intersection of copulas and kernel density estimation (KDE). These include existing implementations in SDV and baselines we constructed, along with a differentially private Gaussian Copula model.

**Copula+KDE Baselines.** The comparison space includes a variety of copula-based and KDE-based extensions. Tables 26 present results on the Adult, Default, Magic, and Beijing datasets.

CopulaGAN applies an empirical copula transform followed by GAN-based generation, while GaussianCopula also relies on the copula transform but pairs it with Gaussian marginals. The GaussianKDECopulaSynthesizer, implemented in SDV with `default_distribution="gaussian_kde"`, fits Gaussian KDEs on marginals but is extremely slow in practice, requiring about 2.5 hours compared to roughly one minute for TabKDE. We also construct a CopulaKDE baseline, which uses an empirical copula transform followed by a $d$-dimensional Gaussian KDE in latent space $[0, 1]^d$ with the bandwidth $\sigma$ set to the median pairwise distance.

Below the lines in Tables 26 are our variants. CopulaDiff represents our diffusion-based variant applied after a copula transform. SimpleKDE is a variant of TabKDE that incorporates a DCR-calibrated kernel and covariance-aware directions, but omits boundary-aware sampling. Finally, TABKDE is our proposed method.

Three central observations arise from the experiments. In terms of accuracy, TabKDE consistently achieves the best or near-best fidelity. While some competitors such as CopulaDiff, SimpleKDE, or GaussianKDECopula perform strongly on C2ST, they typically fall short in marginal or pairwise errors. Regarding privacy, the SDV copula models and the DP Gaussian Copula reach DCR values close to the ideal 50%, but they incur substantially higher errors. As expected, stronger privacy guarantees correlate with reduced fidelity. TabKDE offers a balanced tradeoff, maintaining high fidelity while still achieving moderate privacy with DCR around 58%. Finally, in terms of scalability, GaussianKDECopulaSynthesizer is prohibitively slow, taking several hours compared to TabKDE's single-minute runtime. TabKDE thus emerges as both more efficient and more accurate.

| **Adult** | Marginal Err. ($\downarrow$) | Pairwise Corr. ($\downarrow$) | DCR ($\rightarrow$ 50) | C2ST ($\uparrow$) |
|---|---|---|---|---|
| CopulaGAN | 8.53% | 16.75% | 49.58% | 0.60 |
| GaussianCopula | 12.41% | 19.24% | **50.23%** | 0.18 |
| CopulaKDE | 7.39% | 14.14% | 53.00% | 0.76 |
| GaussianKDECopula (8,461s) | 8.30% | 8.84% | 49.28% | 0.89 |
| CopulaDiff | 2.10% | 4.61% | 50.34% | 0.86 |
| SimpleKDE | 1.98% | 4.64% | 62.89% | 0.90 |
| TabKDE | **1.56%** | **4.51%** | 62.23% | **0.92** |

| **Default** | Marginal Err. ($\downarrow$) | Pairwise Corr. ($\downarrow$) | DCR ($\rightarrow$ 50) | C2ST ($\uparrow$) |
|---|---|---|---|---|
| CopulaGAN | 11.50% | 21.13% | 52.98% | 0.74 |
| GaussianCopula | 12.33% | 21.90% | **49.59%** | 0.41 |
| CopulaKDE | 8.99% | 12.48% | 56.36% | 0.58 |
| GaussianKDECopula (9,239s) | 7.02% | 7.58% | 50.41% | 0.95 |
| CopulaDiff | **1.47%** | **3.29%** | 50.96% | **0.98** |
| SimpleKDE | 3.33% | 5.16% | 66.05% | 0.87 |
| TabKDE | 1.55% | 9.93% | 63.46% | 0.96 |

| **Magic** | Marginal Err. ($\downarrow$) | Pairwise Corr. ($\downarrow$) | DCR ($\rightarrow$ 50) | C2ST ($\uparrow$) |
|---|---|---|---|---|
| CopulaGAN | 10.21% | 9.03% | 51.23% | 0.66 |
| GaussianCopula | 11.19% | 6.34% | **50.10%** | 0.51 |
| CopulaKDE | 11.42% | 7.20% | 55.40% | 0.73 |
| GaussianKDECopula (1,477s) | 2.30% | 5.00% | 50.22% | **0.99** |
| CopulaDiff | 0.94% | **1.72%** | 52.03% | 0.94 |
| SimpleKDE | 3.12% | 3.30% | 62.62% | 0.97 |
| TabKDE | **0.78%** | 2.72% | 63.02% | 0.94 |

| **Beijing** | Marginal Err. ($\downarrow$) | Pairwise Corr. ($\downarrow$) | DCR ($\rightarrow$ 50) | C2ST ($\uparrow$) |
|---|---|---|---|---|
| CopulaGAN | 7.79% | 12.11% | 50.89% | 0.78 |
| GaussianCopula | 10.01% | 6.00% | **50.05%** | 0.11 |
| CopulaKDE | 12.04% | 17.07% | 53.94% | 0.74 |
| GaussianKDECopula (9,288s) | 2.69% | 6.56% | 50.66% | **0.99** |
| CopulaDiff | 2.13% | 4.50% | 50.29% | 0.96 |
| SimpleKDE | 2.06% | 4.68% | 55.45% | 0.94 |
| TabKDE | **1.37%** | **3.74%** | 54.24% | 0.95 |

Table 26: TABKDE vs Copula+KDE Baselines.

## H.2 DIFFERENTIALLY PRIVATE GAUSSIAN COPULA

In addition to the above, we benchmark a differentially private (DP) Gaussian Copula model with a specified privacy budget $\epsilon$. Table 27 reports the results.

Note that while this achieves strong DCR score (as do similar copula methods without DP guarantees), there is not a clear advantage as the $\epsilon$ parameter is decreased. However, the resulting Marginal Error (about 17%) and Pairwise Correlation (about 28%) scores are significantly higher than most other baselines; TABKDE achieves $1.56\%$ and $4.51\%$. Also C2ST (about $0.35$) is worse that most baselines; TABKDE achieves $0.92$, where higher is better.

So while this method does provide a DP guarantee, it appears to perform significantly worse in all accuracy measures, even for very large $\epsilon$ values.

| DP Gaussian Copula (Adult) | Marginal Err. ($\downarrow$) | Pairwise Corr. ($\downarrow$) | DCR ($\rightarrow$ 50) | C2ST ($\uparrow$) |
|---|---|---|---|---|
| $\epsilon = 0.1$ | 18.44% | 32.56% | 49.88% | 0.37 |
| $\epsilon = 1$ | 16.72% | 29.04% | 50.12% | 0.36 |
| $\epsilon = 5$ | 16.27% | 28.69% | 49.97% | 0.37 |
| $\epsilon = 10$ | 16.92% | 27.76% | 49.75% | 0.33 |
| $\epsilon = 100$ | 17.46% | 29.80% | 49.80% | 0.29 |

Table 27: DP Gaussian Copula on Adult with different privacy budget $\epsilon$ (smaller implies stronger privacy).

