# OpenReview forum: "TabKDE:  Simple and Scalable Tabular Data Generation with Kernel Density Estimates"
_ICLR.cc/2026/Conference — ICLR 2026 Conference Withdrawn Submission_

### Official Review · Reviewer_Ka2m · 2025-10-29

**Soundness:** 2
**Presentation:** 2
**Contribution:** 2
**Rating:** 2
**Confidence:** 3

**Summary:**

The paper introduces TABKDE, a three-stage pipeline for tabular synthesis: (1) encode each column (numeric, ordinal, categorical) into a single continuous value, (2) transform the features into a copula-based latent space, and (3) generate new samples via a KDE-style perturbation procedure before inverting back to the original schema. The approach avoids heavy generative training. Experiments on six datasets indicate reasonable distribution preservation and privacy relative to computational cost.

**Strengths:**

**Simplicity and scalability.** The method avoids one-hot encodings, VAEs, and diffusion training; the generation mechanism is simple, efficient and potentially effective scalable to large-scale data.

**Clear end-to-end workflow.** The encode → latent mapping → sample → inverse-copula decode pipeline is clearly presented and straightforward to follow.

**Weaknesses:**

**Writing clarity.** Several sections could be revised for readability and precision.
- Line 166 (categorical values). It appears this treatment may apply only to ordinal columns; please clarify the intended scope.
Metric naming. Line 435: the term “DCR score” is confusing, since DCR is a distance while the “score” is a percentage-based statistic. Consider distinct names.
- Organization. Consider moving most related work to a dedicated section so the introduction can focus on motivation and intuition for the proposed method.
- Related work coverage. The paper should discuss copula-based approaches for synthetic tabular data, such as: Kamthe et al., “Copula Flows for Synthetic Data Generation”; Sun et al., “Learning Vine Copula Models for Synthetic Data Generation”.

**Categorical encoding justification.** Additional analysis is needed:
- [Q1.1] Any numeric encoding of a categorical column imposes an order. What is the impact on model behavior and sample quality?
- [Q1.2] For high-cardinality categoricals, can the model reliably generate the full support, including rare categories?
- [Q1.3] Line 175. If a categorical column is independent of numeric columns but depends on other categorical columns, how is this dependence preserved?
- [Q1.4] Line 327. Total variation distance can underweight long-tail categories. Table 4 mixes numeric and categorical columns, which may obscure behavior. Please report per-column categorical marginals and include TV and KL metrics, with emphasis on rare classes.

**Ablation study needed**
- [Q2.1] How would the proposed encoding method work with existing generative models?
- [Q2.2] How would the KDE model work with one-hot encoding?

**Questions:**

[Q3] Line 157. What theoretical justification or intuition supports the KDE-style generative procedure in latent space?

[Q4] Line 192. Is it fair to define and evaluate DCR in the learned latent space, given that the method is tailored to that geometry? Could this favor the proposed approach over baselines evaluated in other spaces?

[Q5] Line 433. Why is 68 percent considered “serviceable”? Please provide a rationale and, if possible, confidence intervals or statistical context.

---

### Official Review · Reviewer_nGAe · 2025-11-01

**Soundness:** 2
**Presentation:** 3
**Contribution:** 3
**Rating:** 4
**Confidence:** 4

**Summary:**

The paper introduces TabKDE, a simple, scalable, and privacy-aware framework for generating good quality synthetic tabular data by carefully combining classical statistical tools (namely, copula transformations and kernel density estimation) without relying on complex deep generative models such as GANs, VAEs, or diffusion models. The approach maps mixed-type tabular data into a numerical encoding which is then transformed to a copula-based latent space, where KDE-based sampling of new synthetic latent data takes place. The synthetic latent values are then transformed back to synthetic mixed data via an inverse copula transformation followed by a decoding step.  While TabKDE does not achieve SOTA performance with regard to fidelity and privacy metrics it still performs reasonably well and is considerably more scalable than the more complex deep generative models.

**Strengths:**

The paper proposes an interesting new framework for generating synthetic tabular data. (It is refreshing to see work outside of deep generative models.)

While none of the components used by TabKDE is new, the paper combines these components in a clever and neat way.

In addition to comparing TabKDE to popular baselines in the field, the paper also perform comparisons to several hybrid models that mix elements of TabKDE with diffusion and VAE models.

**Weaknesses:**

The paper presents a combination of evaluation results obtained both from its own experiments and from prior publications (specifically Zhang et al., 2024, and Shi et al., 2025). However, some of these results are not directly comparable because different data splits were used.

As stated in lines 972–976 of the paper:

“We consider two ways to split data into test and train set. The numbers in Table 9 reflects the splits done by TABSYN, which we maintain for direct comparison. This split was not even in size, partially to ensure there were no categories in the Test split which were not present in the Train split. When we do not directly compare to results in the Zhang et al. (2024), we use a different random and even split.”

Despite this distinction, the $\alpha$-precision and $\beta$-recall results in Tables 22 and 23 include values extracted from Zhang et al., even though the captions indicate that the results for TabKDE (and several other baselines) were obtained using equal-sized splits. This means the reported comparisons are not strictly valid, as they mix results based on incompatible data partitioning schemes. To ensure fair and meaningful comparisons, the authors should either re-run the TabKDE and other baseline evaluations using the original Zhang et al. splits, or recompute all baseline results from Zhang et al. under the equal-sized split protocol.


Additional suggestions for improvement:

Sometimes the paper runs its own reproduction of the SMOTE and TabSyn baselines and sometimes it does not. For instance, the results for the marginal distribution alignment (Table 4), pairwise correlation alignment (Table 5), C2ST (Table 7), and DCR (Table 9) metrics are based on the paper’s own reproduction while for $\alpha$-precision (Table 22) and $\beta$-recall (Table 23) it is not.  The paper should fix these discrepancies.

The list of baselines is not consistent across all metrics. For instance, the paper compares against TabDiff in Table 8 but not in Tables 4, 5, and 7. These missing comparisons should be included. Furthermore, some baselines are only presented in the Appendix tables but not in the main text ones. The paper should harmonize (i.e., have the same set of baselines) across all the results.

The paper report in Tables 15, 17, and 20, average results and standard deviations (over 10 runs) only for the TabKDE generator and only for a few selected evaluation metrics. But it should report averages and SDs across 10 runs for all metrics and baseline models.

If possible, the paper should consider increasing the number of benchmark datasets.

Minor suggestions:

The style of the citations is sometimes awkward. For instance, in line 40 the citation “Copula-based data generators Patki et al. (2016)” should be changed to “Copula-based data generators (Patki et al., 2016)”.

Typos:

Line 21: “larger than prior art” -> “larger than prior state-of-the-art” ?

Line 35: “need to paired” -> “need to be paired”

Line 60: “over come” -> “overcome”

Line 66: “categorical values” -> “ordinal values”

Line 142: “sample” -> “sampled”

Line 160: “at randomly” -> “at random”

Line 194: “can comparing” -> “can compare”

Line 239: line 6 of algo 3 is missing parenthesis

Line 296: “Noteably” -> “Notably”

Line 377: Table 6 is misplaced

Line 666: “at randomly” -> “at random”

Line 676: “This means, do not” -> “This means, we do not”

Line 780: “can mapped back” -> “can be mapped back”

Line 791: “)}” -> “})”

Line 793: “$z_ij$” -> “$z_{ij}$”

Line 821: “computes” -> “compute”

Line 860: “define out kernel” -> “define our kernel”

Line 871: “And Moreover” -> “And moreover”

Line 872: “beleive” -> “believe”


INITIAL ASSESSMENT:

Although TabKDE does not achieve state-of-the-art performance, its strong scalability offers a practical and compelling trade-off. Moreover, the paper contributes a fresh perspective to a field largely dominated by deep generative models.

I am assigning a relatively low score at this stage due to the pending clarifications regarding the experimental evaluations. However, I believe these issues can be readily addressed, and I would be happy to raise my score once satisfactory clarifications are provided.

**Questions:**

In addition to Tables 22 and 23, are there any other results generated with the alternative equal sized data splits?

For the comparisons with Zhang et al., did the paper, in addition to using the same data splits, also replicate other aspects of that refercence's evaluation pipeline (such as data preprocessing steps and related procedures) to ensure the results are comparable?

---

### Official Review · Reviewer_sXrU · 2025-11-01

**Soundness:** 2
**Presentation:** 1
**Contribution:** 2
**Rating:** 2
**Confidence:** 4

**Summary:**

This paper proposes a scalable and simple tabular data generation method using KDE.
The proposed method, TabKDE, first encodes raw data into a Euclidean space.
To avoid one-hot encoding for categorical features, they apply principal vector-guided encoding, which projects each category along the main principal component direction of the numerical features, preserving semantic relationships between categories.
Then, the embedded data are mapped into a continuous latent space $[0,1]^d$ using a copula transform, which normalizes each feature while maintaining inter-feature dependencies.

For sampling, they use a KDE-based generative process, where a training point in the latent space is randomly selected, a direction proportional to the data covariance is chosen, and a new point is generated by adding a small stochastic deviation. Out-of-bound points are re-sampled to stay within $[0,1]^d$, ensuring the validity of marginal distributions.

The authors also propose a coreset construction method that selects a small representative subset of the data to approximate the same density structure, improving scalability and computational efficiency.

They claim that the proposed method outperforms several baselines in multiple aspects, including scalability, efficiency, accuracy, and privacy.

**Strengths:**

The idea of constructing a coreset to approximate the KDE distribution is simple yet fresh, effectively improving scalability without significant loss of accuracy.

**Weaknesses:**

- $\textbf{Inconsistent Baselines}$:
The comparisons are not standardized --  some baselines are re-implemented, others are taken from prior work, and several (e.g., GReaT, GOGGLE) lack reported metrics, making fair evaluation difficult.

- $\textbf{Unclear Privacy Criteria}$:
The marks in Table 1 are not based on consistent quantitative thresholds.
Models with near-ideal DCR values (e.g., CoDi, STaSy) are marked x, while others with higher DCR (e.g., TabDDPM) are v without clear justification.

- $\textbf{Lack of Efficiency Evaluation}$:
Despite claiming “simplicity and scalability,” the paper provides no quantitative metrics such as training time, sampling speed, or memory footprint. Scalability is discussed only qualitatively.


- $\textbf{Limited Empirical Analysis}$:
The paper lacks ablation studies or visualizations to isolate the impact of each component (copula transform, coreset construction, etc.).


- $\textbf{Presentation and Formatting Issues}$:
The presentation quality is uneven: several tables exceed page margins, algorithm listings are poorly aligned, and figure/table placement breaks text flow.
These layout inconsistencies make the paper harder to follow and reduce readability.

**Questions:**

1. When sampling new points from the KDE, how are categorical features mapped back to discrete categories during the inverse copula transform?


2. Since the KDE samples may slightly move outside the $[0, 1]^d$ bounds, how does the method ensure that categorical mappings and marginal ranges remain valid after inversion?

**Details Of Ethics Concerns:**

no ethic concerns

---

### Official Review · Reviewer_kD6Y · 2025-11-19

**Soundness:** 1
**Presentation:** 1
**Contribution:** 1
**Rating:** 0
**Confidence:** 5

**Summary:**

-

**Strengths:**

--

**Weaknesses:**

--

**Questions:**

The authors violated the margins as defined by ICLR 2026 template at page 7 hence my recommendation is desk reject

---

### Note · Authors · 2025-11-19

I have read and agree with the venue's withdrawal policy on behalf of myself and my co-authors.